

# Machine learning nowcasting of the Vögelsberg deep-seated landslide: why predicting slow deformation is not so easy

Adriaan van Natijne[1], Thom Bogaard[2], Thomas Zieher[3], Jan Pfeiffer[3], and Roderik Lindenbergh[1]

[1]Department of Geoscience & Remote Sensing, Faculty of Civil Engineering and Geosciences, Delft University of Technology, Delft, the Netherlands
[2]Department of Water Management, Faculty of Civil Engineering and Geosciences, Delft University of Technology, Delft, the Netherlands
[3]Institute for Interdisciplinary Mountain Research, Austrian Academy of Sciences, Innrain 25, 6020 Innsbruck, Austria

**Correspondence:** Adriaan van Natijne (A.L.vanNatijne@tudelft.nl)

**Abstract.**

Landslides are one of the major weather related geohazards. To assess their potential impact and design mitigation solutions, a detailed understanding of the slope processes is required. Landslide modelling is typically based on data-rich geomechanical models. Recently, machine learning has shown promising results in modelling a variety of processes. Furthermore, slope
conditions are now also monitored from space, in wide-area repeat surveys from satellites. In the present study we tested if use of machine learning, combined with readily-available remote sensing data, allows us to build a deformation nowcasting model. A successful landslide deformation nowcast, based on remote sensing data and machine learning, would demonstrate effective understanding of the slope processes, even in the absence of physical modelling. We tested our methodology on the Vögelsberg, a deep-seated landslide near Innsbruck, Austria. Our results show that the formulation of such machine learning system is not
as straightforward as often hoped for. Primary issue is the freedom of the model compared to the number of acceleration events in the time series available for training, as well as inherent limitations of the standard quality metrics. Satellite remote sensing has the potential to provide longer time series, over wide areas. However, although longer time series of deformation and slope conditions are clearly beneficial for machine learning based analyses, the present study shows the importance of the training data quality but also that this technique is mostly applicable to the well-monitored, more dynamic deforming landslides.

## 1 Introduction

Landslides make up for 6% of the weather related disasters globally (WMO, 2019). To protect the public, landslides have been a major research topic for the last decades. For local landslide mitigation by geotechnical intervention an up-to-date understanding of these hydro-meteorological phenomena, their feedbacks and impact is desired. This understanding may then be leveraged for the design of landslide hazard mitigation measures.



Where the installation of effective remediation concepts is not possible, Early Warning Systems may help to reduce the land-
slide risk. Such system should quickly adapt to changing conditions, both on the slope and global (e.g. climate change).
Moreover, such a system should be fast to adapt and implement to assess as many slopes as possible.

Existing local systems typically provide early warning based on in-situ slope monitoring (Guzzetti et al., 2020). An example
of a satellite based, global Early Warning System is the LHASA model (Kirschbaum and Stanley, 2018; Hartke et al., 2020;
Stanley et al., 2021) that provides a global nowcast of acute landslide susceptibility. However, these systems typically focus on
sudden, fast, and shallow landslides. Such catastrophic events change the landscape, and as a consequence the situation before
and after the collapse are no longer comparable. Therefore, the landslide process preceding the collapse can only be studied if
data from before the landslide is available.

We focus on slow moving, reactivating, deep-seated landslides on natural slopes, for which the deformation pattern is controlled
by hydro-meteorological forcing. These deep-seated landslides are estimated to comprise 50% of the landslides globally (Her-
rera et al., 2018; Novellino et al., 2021). The deep-seated landslides we focus on rarely evolve into catastrophic collapse and
often entail a complex response to hydro-meteorological conditions controlling the landslide's pore pressure (Bogaard and
Greco, 2015). They are characterised by gradual, non catastrophic, deformations that can be responsible for extensive infras-
tructure damage (Mansour et al., 2011). Deformation rates typically vary from millimeters to decimeters per year, whereas
phases of acceleration or deceleration often correlate time-delayed with hydrological conditions (Intrieri et al., 2018).

Monitoring systems only supported by the detection of currently emerging acceleration events (e.g. Carlà et al., 2017), may only
be used to detect already ongoing acceleration. As a consequence, adequate early warning is only possible if the deformation
can accurately be predicted beforehand. Therefore, the deformation should be predicted from the predisposing conditions on
the slope, combined with dynamic factors such as infiltrating precipitation and snowmelt that lead to higher pore pressures,
instability and subsequent deformation. However, the deformation behaviour of such slow, deep-seated landslides is 'extremely
difficult' to model (Van Asch et al., 2007).

Past landslide deformation events are indicative of the future behaviour, as landslides are likely to display similar behaviour
in similar situations (Fell et al., 2008; Guzzetti et al., 1999). Unlike to catastrophic landslides, where the landslide dynamics
change permanently, slow moving landslides are not single, catastrophic incidents. Therefore, analysis of the monitoring data
of deep-seated landslides are expected to reveal causal factors in landslide deformation, which allow for a continuous cycle of
forecast and validation of the relation between deformation and the conditions on the slope.

Deformation nowcasting could be considered an intermediate option between monitoring and modelling, integrating sensor
data to estimate the current situation (the system state) and extrapolate on a short timescale. New data and data integra-
tion methods, 'machine learning', offer new possibilities for such data-driven landslide forecasting (van Natijne et al., 2020).
Furthermore, these techniques offer new capabilities to continuously track the system state without extensive, in-situ sensor
networks and physics-based modeling. Such data-driven model will 'learn' the landslide dynamics and the interplay of hydro-
meteorological factors from the deformation signal of the landslide.





In the last decades satellite observations have increased in quantity, shortening the time between subsequent acquisitions, as well as increasing the variables observed (Belward and Skøien, 2015). These acquisitions provide us with a global overview of the status of the earth at local scale, often with weekly to daily updates. More recently there is the tendency to make the data freely available, a development that lowered the barrier for innovations (Zhu et al., 2019), and especially benefits experiments that require long time series, like this study. Even though their coverage is often limited to the surface, the repeated monitoring of the slope conditions may reveal the slope processes (van Natijne et al., 2020).

Here we present a data-driven nowcasting model with a four day lead time of the deformation of the Vögelsberg landslide, near Innsbruck, Austria. We use readily available, remotely sensed, data and products, and test various similar remote sensing products to assess their relative performance in the nowcasting model. We discuss the complications encountered during modelling: over-parametrization, the impact of optimization metrics, and the challenges due to the deep-seated landslide inertia compared to the highly dynamic forcing of the slope.

First, we introduce the modelling options, and study area. Second, we present the resources available to us, and our modelling approach, followed by the results and an extensive discussion on the insights gained during the modelling exercise. Last, we provide recommendations for future data-driven landslide nowcasting exercises.

## 2 Data-driven modelling approaches

In the present study we interpret data-driven modeling as a form of naive modelling, that is unaware of the physics behind the landslide process. For data-driven models, the deformation of the slope is merely a signal to be reproduced from a collection of observations by empirical relations, in contrast to traditional, landslide geomechanical modelling, that is rooted in physics. In recent years, data-driven landslide deformation nowcasting has gained popularity, as illustrated by the abundance of studies in Table C.1. Various examples come from landslides around the Three Gorges Dam that are strongly controlled by the reservoir water level. However, this is not the most common type of deep-seated landslide, where instead deformation is driven by the water storage in the deeper subsurface controlled by a long-term water balance of precipitation and snowmelt input, evaporation losses and regional groundwater input and drainage (Bogaard and Greco, 2015).

Therefore, indirect transfer from precipitation and snowmelt to storage has to be captured. For example, by including recent observations in a bucket model (Nie et al., 2017), that typically simplifies the subsoil as a storage that is replenished by precipitation and emptied by drainage and evaporation. Furthermore, changes to the storage may involve a time delay, depending on complex infiltration processes. This process may be dependent on the precipitation type, duration and intensity. Moreover, deformation may not be governed by a short and single precipitation event. For example, a short, extreme precipitation event or three days of consecutive drizzle may introduce similar amounts of water to the system, but will be represented differently in storage changes due to different infiltration abilities of the soil. All in all, modelling of deep-seated landslides will likely require some form of storage modelling, where these dynamics are either resolved by the model or in advance by an expert.





Two distinct modelling approaches can be distinguished. Modelling is either based on classification of the environmental
conditions and associated deformation response, or calculates the expected deformation response from the conditions on the
slope. In either case the model parameters are tuned on historic observations such that they best reproduce the deformation
signal from the conditions observed previously at the slope. Both will be introduced briefly.

## 2.1 Classification methods

Based on the assumption that similar conditions trigger a comparable deformation response (Fell et al., 2008; Guzzetti et al.,
1999), conditions and responses may be categorized. The current slope conditions are then matched against historic conditions,
and the deformation response is assumed to be the same. Extrapolation of the response to previously unencountered conditions
is typically impossible via this method. However, the system will therefore also not yield unrealistic results, and could be
considered bound to the previously encountered deformation signal.

## 2.2 Continuous models

The simplest, linear, model is the weighted sum of the quantified conditions at the slope. However, the slope response may not
be linear and is typically not instant. Neural networks make may be used to estimate any signal by the formation of a network
of interlinked nodes that ingest and combine the conditions on the slope in subsequent layers of nodes (Hornik et al., 1989;
Hill et al., 1994). A time series passed to a single input neuron is equal to a weighted sum of the time series, plus a bias.

As more hidden layers of neurons are introduced to the system, the direct link to the (time series) input is lost, as combinations
are made. Furthermore, an activation function may be applied to scale the output of each node, especially to normalize the
response and filter outliers, at the cost of introducing non-linearity to the system. The number of parameters, degrees of
freedom of the model, are associated with the number of input variables. When historic observations are supplied as additional
observations, they will each require their own model parameters, and increase the degrees of freedom in the model.

State aware models, such as Recurrent Neural Networks (Connor et al., 1994), maintain a track record of the state of the
landslide instead, and iterate over the input time series in successive model runs. Individual observations are fed into the system,
with the system maintaining track of their contribution to the current state of the landslide. These models resemble a bucket
model, a simplified representation of the water storage in the subsoil. However, unlike in a traditional (soil moisture/ground
water) bucket model, all variables are taken into account, even if they do not directly represent water. Furthermore, unlike
regular neural networks, the number of trainable parameters is not dependent on the length of the history supplied to the
model, but on the number of memory cells and time series.

Models based on Recurrent Neural Networks suffer from computational difficulties during optimisation, where gradients may
vanish (Bengio et al., 1994; Hochreiter and Schmidhuber, 1997; Hochreiter, 1998). Therefore, they are typically replaced by
models based on Long Short-Term Memory (LSTM) nodes (Hochreiter and Schmidhuber, 1997), that do not suffer from this
due to built-in normalisation. Each LSTM 'bucket' is capable of weighting, retaining and clearing a memory of previous inputs,
and as such tracks the system state.





The challenge specific to forecasting and nowcasting is the absence of information on the future slope conditions. The latest information available to the system are the current conditions and the last estimation of the system state. Auto-regressive models predict these conditions as well, so that subsequent forecasts may use these environmental conditions in their models. However, especially precipitation is governed by external influences and may not be predictable from the other forcing parameters in the system. As an alternative, forecasts may be included into the model. However, this would require forecasts for all input variables. Therefore, such system was deemed not suitable for this application.

Special attention should be paid to the robustness of the model. Even ten years of daily observations will result in less a time series of less than 4 000 epochs, much less than desirable for use in machine learning (Cerqueira et al., 2019). If too few training data is provided, the abundance of input data creates unique combinations of conditions and outputs. This will lead to excellent performance during training, but reduced performance during testing and application, and is known as over-fitting.

There are infinite data-driven modelling possibilities and the generic character of many data-driven models suits the diversity in available remote sensing variables. However, due to the limited length of the time series, in comparison to typical machine learning studies, one should stay close to the physics and processes, to limit the freedom of the model towards a solution. Therefore, one has to ensure a balance between the number of parameters to be estimated and the training/validation data available.

## 3 Case study: the Vögelsberg

The Vögelsberg is a deep-seated landslide, located in the Wattens basin, near Innsbruck, Austria (Figure 1). Its north-east facing slope covers approximately 4.6 km$^2$, and ranges between 750 m and 2200 m above sea level. A nearby weather station reports an average yearly precipitation of 896 mm, of which 13% is in the form of snow. The lower, active part of the landslide is only about 0.2 km$^2$ and is covered by pasture fields, sparse forests and few houses and farm buildings. The shearzone was identified via inclinometer measurements to be at 43–51 m below the surface, although strongly disintegrated soil up to 52–70 m indicates a long history of activity (Pfeiffer et al., 2021).

In 2016 an Automated Total Station (ATS) was installed in Wattenberg, opposite to Vögelsberg, by the Division of Geoinformation of the Federal State of Tyrol. The system surveyed each of the fifty-three benchmarks every hour. Extensive corrections to the measurements were necessary, due to the instability of the monument the total station is located on, as well as atmospheric disturbances due to the Alpine conditions. Measurements were provided by the Division of Geoinformation of Tirol as a series of pre-processed range measurements relative to the start of the measurements. The time series of the displacement rate at the two benchmarks are shown in Figure 2.

The deformation of the Vögelsberg landslide is a complex response to the hydro-meteorological conditions in the catchment, in particular precipitation and (delayed) infiltration from snowmelt. A binary prediction of stability/instability or acceleration/deceleration is insufficient for the Vögelsberg, as the slope is undergoing continuous deformation. Pfeiffer et al. (2021) conducted a full assessment of the hydro-meteorological drivers and found a 20–60 day time lag between rainfall and acceleration and




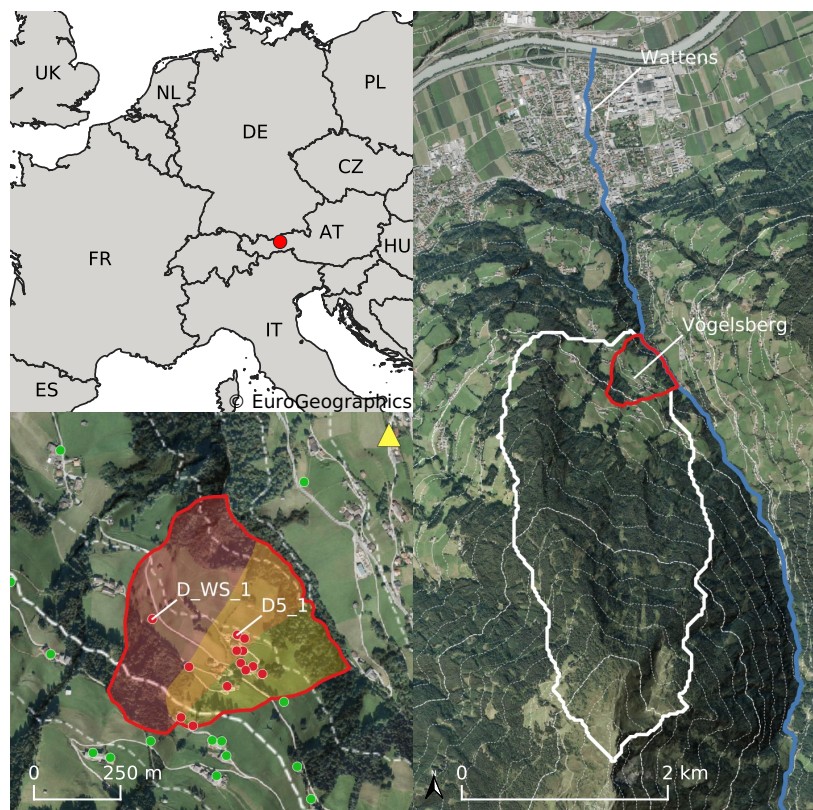

**Figure 1.** Overview of the landslide catchment (white) and active region of the Vögelsberg landslide (red). The northern subsection of the slope (red) and southern (yellow) section and overlapping area are marked in the detail map. Dashed contour lines are shown every 100 meters of elevation change. Out of a total 53 retroreflecting prisms, the 29 benchmarks with the longest time series (2016–2020) are shown. Benchmarks on the landslide are shown in red, stable, reference benchmarks in green. The time series of benchmarks 'D_WS_1' and 'D5_1' are shown in Figure 2. The location of the total station on the opposite slope is marked by a yellow triangle. (Backgrounds: Eurostat/EuroGeoGraphics; Federal State of Tyrol, Austria)

a 0–8 day time lag between snowmelt and acceleration. Noteworthy is the difference in behaviour between the northern and southern sections of the slope, represented by benchmarks 'D_WS_1' and 'D5_1' respectively (Figure 2). The northern section

of the slope ('D_WS_1') shows a higher variability in the deformation signal, with stronger accelerations than the southern, inhabited, section of the slope ('D5_1'). We focus on these two benchmarks, as a balanced representation of the two landslide sub-systems.

The deformation rate, derived from the total station range measurements, was smoothed by a moving average filter till few, noise induced, negative (up-slope) deformations remained, while maintaining the highest possible temporal resolution (Fig-

ure B.1). A moving average of the most recent 32 days was necessary to remove most of the noise. As a consequence, the onset of acceleration will be only 1/32 of the signal, stressing the need for an acceleration prediction rather than extrapolation



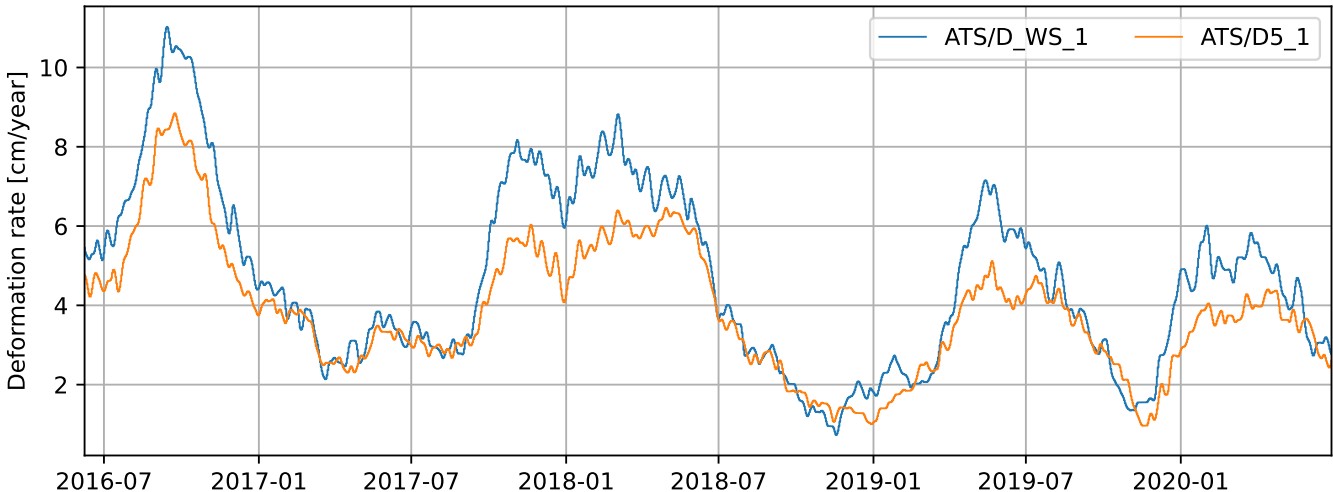

**Figure 2.** Daily deformation rate of the Vögelsberg at benchmarks 'D5_1' and 'D_WS_1' (Figure 1), as measured by the automated total station and smoothed by a moving average filter over the last 32-days.

of deformation measurements as warning signal. Furthermore, the amplitude of the filtered signal lags behind the original deformation signal, as only historic observations may be used in an operational system. Moreover, signals shorter than the filter length will be reduced in amplitude.

## 4 Methodology

Our model aims to predict the landslide deformation, based solely on the current conditions at the slope. No (precursory) deformation data or prior defined geomechanical model is used during prediction. The main model constraints are that we have a relatively limited amount of data points (1 482 samples) and will work with readily available remote sensing data and products. Furthermore, we set the objective to model with daily time steps and a forecast lead time of 4 days.

With these constraints in mind, a system was designed based on a parsimonious recurrent neural network. First, we will introduce the data available. Second, an overview is provided of the pre-processing applied to the input variables. Third, we provide the specifications of our model. Last, the training and validation of the model are discussed.

### 4.1 Model variables

The model variable selection is based on the analysis of factors of influence (Pfeiffer et al., 2021), and are mainly of data-driven nature. Pre-disposing or causal factors, such as topography, that are necessary for a landslide to form, are considered static in this study. Therefore, the focus is on the dynamic conditions leading up to landslide instability and deformation, and triggering factors. The selection of variables is listed in Table 1.





Our method is designed with the intent to be generally applicable. Therefore, where possible, remote sensing products were used, as they are likely to be available elsewhere as well. Where available, redundant products, that represent the same or

similar quantities, were included to assess their relative performance in the nowcasting model. The correlation between the products is limited (Figure A.1), indicating differences between the products of the same quantity. Effects that may not be observed directly, such as soil moisture under snow, require some form of modelling or re-analysis. These quantities, not directly available from remote sensing, are taken from re-analysis models 'ECMWF Re-Analysis, version 5' (ERA5) and the 'Global Land Evaporation Amsterdam Model' (GLEAM).

The desired output of our model is a daily, four days ahead prediction of the landslide deformation rate at benchmarks 'D_WS_1' and 'D5_1'. Reference, training and validation samples are provided by the automated total station located on the Wattenberg, opposite to the Vögelsberg (Figure 1). Deformation measurements were performed hourly from 2016-05-04 to 2020-06-28, and aggregated to 1 482 daily averages to reduce noise. The noise in the signal was further reduced by a 32-day moving average filter, of which the results are shown in Figure 2. The time series at the 51 other benchmarks (Figure 1) were

not used in the modelling.

Daily precipitation information is provided by the Integrated Multi-satellitE Retrievals for GPM (IMERG) algorithm of the Global Precipitation Measurement mission (GPM) (NASA, 2018). 'Early' results are provided with sub-day delay, and are therefore especially suitable for an operational nowcasting model. For comparison daily precipitation from the ECMWF ERA5 Land re-analysis is included as well (ERA5, 2019). Snow properties are covered by two products of the ERA5 Land re-analysis:

snow water equivalent, and snowmelt.

Soil moisture, especially at depth, cannot be observed directly from space at a high enough resolution for this application. The operational products from the Copernicus Land Service, Soil Water Index and Surface Soil Moisture, are frequently unavailable either due to unfavourable slope topography or due to snow cover. Alternatives are provided by SMAP (Entekhabi et al., 2010); a re-analysis from 'Global Land Evaporation Amsterdam Model' (GLEAM) (Martens et al., 2017; Miralles et al., 2011); and

ERA5 Land (ERA5, 2019). Evaporation estimates are taken from GLEAM as well. Air temperature, a proxy indicator of evaporation and snowmelt, is included from ERA5 Land (ERA5, 2019).

## 4.2 Variable preparation

The model is fed with the eleven variables defined in §4.1 (Table 1). Except for the deformation time series, all sources consist of gridded products, with wide area coverage. In this study only the data point closest to the Vögelsberg was used. To match the

time resolution of the deformation measurements the model is run at daily intervals. Observations available at shorter intervals are aggregated to daily means first. Where data is missing, for example due to sensor failure, the values are filled with the data from the previous day (forward filling), as would be possible in an operational scenario. Furthermore, two modelled time series were added to the system: an antecedent precipitation index (API) as basic hydrological model and a random, seasonal noise signal.







**Figure 3.** Overview of the variable space (Table 1). The values are offset to a zero mean and scaled by their standard deviation. A single iteration of the seasonal noise (`fake/fake`) is shown as an example.



**Table 1.** Selection of time series considered for integration into the model. Deformation variables are marked 'D', while slope conditions, input variables to the model, are marked 'V'. Observations are marked 'S' for directly observed variables processed and available within the time frame of a nowcasting system; 'R' for re-analysis variables, and 'M' for variables modelled within this study (see §4.2). References to the various sources are provided in the main text. The internal identification is derived from the variable as referenced by the source, and is used throughout the figures to refer to the various time series. From rasterized products, only the time series closest to the Vögelsberg was used.

|     | Variable | Source | Type | Spatial res. | Temp. res. | Int. identification |
|-----|----------|--------|------|--------------|------------|---------------------|
| D1 | Deformation 'D_WS_1' | ATS (local) | S | point | daily | `ATS/D_WS_1` |
| D2 | Deformation 'D5_1' | ATS (local) | S | point | daily | `ATS/D5_1` |
| V1 | Precipitation | ERA5 | R | $0.1°$ ($\simeq 10$ km) | hourly | `ERA5/tp` |
| V2 | Precipitation | GPM | S | $0.1°$ ($\simeq 10$ km) | 30 min. | `GPM/precipitationCal` |
| V3 | Snow water equivalent | ERA5 | R | $0.1°$ ($\simeq 10$ km) | hourly | `ERA5/swe` |
| V4 | Snowmelt | ERA5 | R | $0.1°$ ($\simeq 10$ km) | hourly | `ERA5/smlt` |
| V5 | Soil moisture, full profile | SMAP | S | $0.1°$ ($\simeq 10$ km) | 3 hrs. | `SMAP/sm_profile` |
| V6 | Soil moisture, root zone | GLEAM | R | $0.25°$($\simeq 25$ km) | daily | `GLEAM/SMroot` |
| V7 | Soil moisture, 100–289 cm | ERA5 | R | $0.1°$ ($\simeq 10$ km) | hourly | `ERA5/swvl4` |
| V8 | Evaporation | GLEAM | R | $0.25°$($\simeq 25$ km) | daily | `GLEAM/E` |
| V9 | Air temperature | ERA5 | R | $0.1°$ ($\simeq 10$ km) | hourly | `ERA5/t2m` |
| V10 | API | | M | point | daily | `API/API` |
| V11 | Sesonal noise | | M | point | daily | `fake/fake` |

The Antecedent Precipitation Index (API, `API/API`, V10) was designed to estimate the water present in the watershed (Kohler and Linsley, 1951; Heggen, 2001). The API is included to determine if such variable could support the model. The parameters were chosen based on trial and error. Precipitation less than 0.1 mm was ignored, in addition a 10% direct evaporation loss, and a 4% daily storage loss is assumed. That is, the API at time step $t$ is calculated as

$$\text{API}_t = \max(0, p - 0.1) \cdot 0.9 + 0.96 \cdot \text{API}_{t-1}, \tag{1}$$

with $p$ the daily precipitation sum. The API is shown in Figure 3.

A random variable with seasonal characteristics is added to the variable selection to analyze the effect of spurious correlation on the model. The random variable, `fake/fake` (V11), based on Brownian motion, is tuned to match a typical seasonal characteristic in the 32-day history relevant to the model. The auto-correlation behaviour is illustrated in Figure 4, and closely resembles the dynamics of the surface temperature as provided by ERA5 for the first 2–3 months. Longer correlation periods

are not relevant for our model.




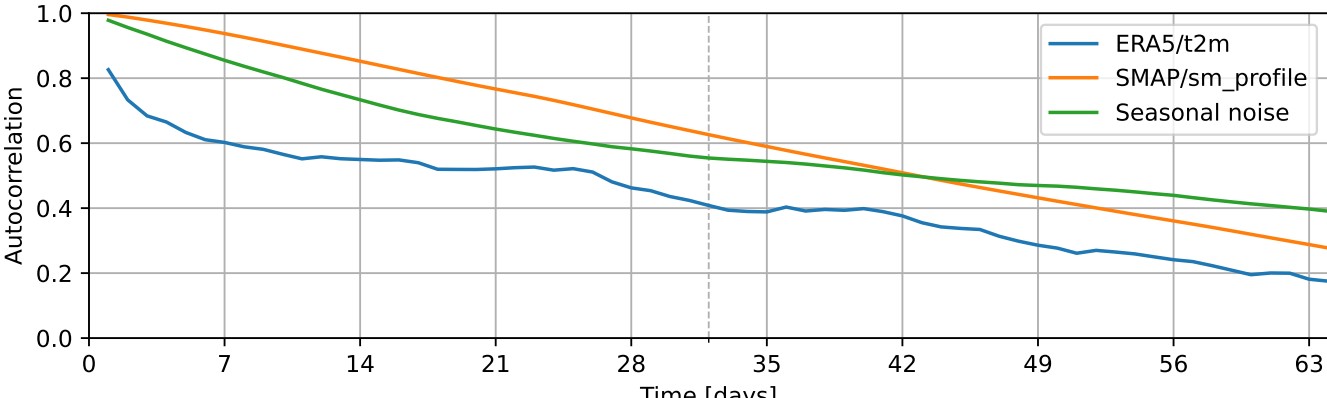

**Figure 4.** Autocorrelation of one of the generated signals compared to the autocorrelation of the temperature as taken from ERA5 and the soil moisture estimate from SMAP. The length of the history as used by the model, 32 days, is indicated by the dashed line.

All variables are offset to become zero-mean and scaled by the standard deviation. Therefore, all input variables are on approximately equal scale and represented as deviations from their average condition. The normalization parameters, mean and standard deviation, should be kept fixed while new data is added, as not to disturb the model. The data set is fed to the model as a time stamped collection of daily observations, illustrated in Figure 3.

### 4.3 Model configuration

Our model is based on a shallow neural network around a single Long Short-Term Memory (LSTM) node (Hochreiter and Schmidhuber, 1997), that resembles a bucket model for the water storage in the subsoil. The model is supplied with a thirty-two day history of observations, equal to the length of the moving average filter, longer than the lag time for snow (0–8 days) and sufficient to cover most of the 20–60 day lag time for rainfall at the Vögelsberg found by Pfeiffer et al. (2021). From a pre-defined, optimized initialisation, the model is cycled for each day of preceding observations, feeding the observations into memory, before a prediction is made based on the final bucket values ($\mathbf{m}$). The model is illustrated in Figure 5, as function of environmental conditions ($\mathbf{x}$, Table 1), at each of the $n = 32$ days preceding the nowcast, the LSTM node and four neurons of a single benchmark, one for each prediction day. This last, output, layer is repeated for both benchmarks ('D_WS_1' and 'D5_1') to be predicted, while the LSTM memory ($\mathbf{m}$) is shared between the benchmarks.

In total, for a network configuration with a single memory cell ($\mathbf{m}$), 68 parameters have to be estimated. The LSTM node, with one hidden state, requires 52 parameters to be estimated for the eleven variables (Table 1). Sixteen parameters are required for the output, eight for each time series: per day one bias and one scaling parameter for the final state of the LSTM node. The number of parameters to be estimated is independent of the history length.

Four parameters are added per extra prediction day (two benchmarks, one bias and weight each). An extra memory cell requires $8h + 4x + 1$ extra parameters, with $h$ the current number of hidden nodes and $x$ the number of input variables. While only four



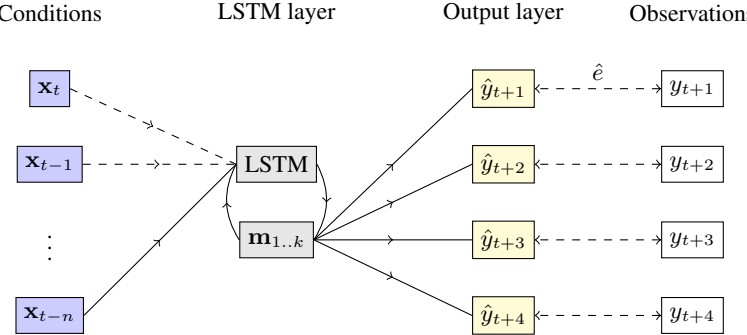

**Figure 5.** Simplified schematic of the model. From left to right: the hydro-meteorological conditions ($\mathbf{x}_t$) on the slope at the current ($t$) and $n$ preceding time steps; the LSTM layer, including its internal feedback and memory cells ($\mathbf{m}_{1..k}$); the output layer $\hat{y}_t$, which combines the $k$ memory-cells $\mathbf{m}$ of the LSTM node to four predictions; the observations $y_t$, as available for comparison during training and validation. During initialisation, the conditions on the slope are fed to the system on a day-by-day basis, starting at the oldest observations. The output layer is only invoked at the last iteration, with the final values of the LSTM memory. The parameters of the LSTM layer are optimized on both deformation time series in parallel, the output nodes are tuned individually for each benchmark.

parameters are added for each additional input variable. Hence, extra memory always requires more parameters than extra input variables.

An interpretation of the network is that the development of the slope state in the last 32-days is described by the LSTM node. The state is scaled, and otherwise matched to the individual benchmarks, by the output neurons. The four days are an extrapolation of the current state of the system, no prediction of the conditions on the slope is made.

The 'mean squared error' was chosen as the loss function. This function, that quantifies the difference between the predicted and observed deformation, is to be minimized during training. The quality of the prediction is measured on the period not used for training. This function assures the cumulative deformation over time is realistic, as errors are balanced between over- and underestimation. Therefore, the predictions will not show a bias towards acceleration or deceleration.

The TensorFlow machine learning framework was chosen to implement the model (Google, 2022). The LSTM model is implemented in a stateless fashion: the warm-up phase is repeated for every nowcast. The model was run on a workstation based on an Intel Xeon W-2123 (4 cores, 8 threads, 3.6 GHz) with 32 GB RAM, while model variations were tested on the high performance computing cluster of the Delft University of Technology. Given the limited size of the region of interest, as well as the limited number of parameters, the full model fits into 1 GB of memory.

## 4.4 Model training & validation

During training the model parameters are tuned such that the final model state best describes the deformation prediction. The model is optimized with the Adam optimizer (Kingma and Ba, 2017). The model is trained on the loss, after 50 training passes





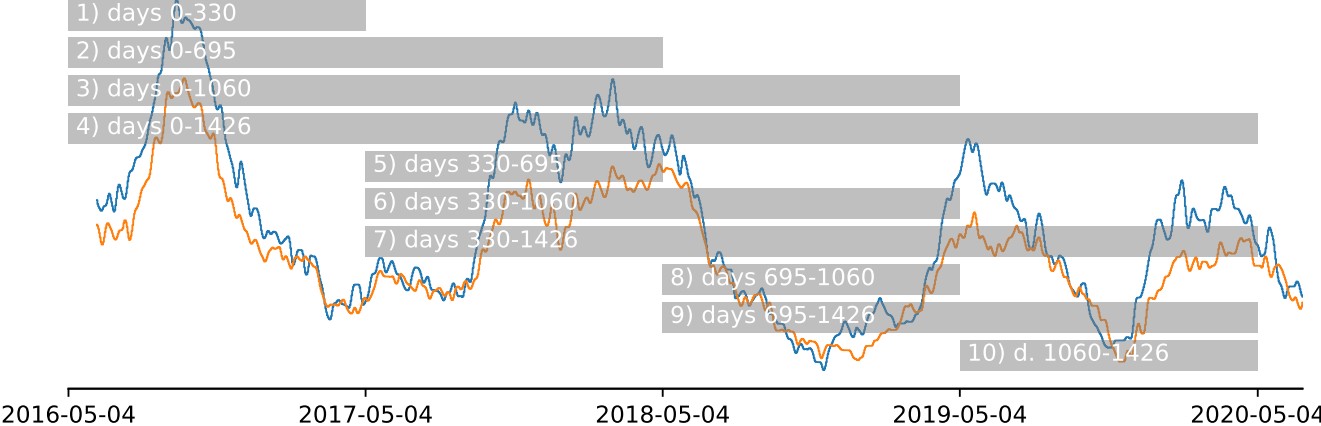

**Figure 6.** Training periods as supplied to the model. The data outside the training period is used for validation. Note that with the longest training period (4), there is very limited validation data left. The deformation pattern (Figure 2) is shown in the background for reference. As there is no clear seasonality in the deformation signal, the data set was split in approximate years from the start of the measurements.

that do not lower the training loss, model parameters are fixed. If this steady state is not achieved after 25 000 passes, the training is stopped anyway and the model parameters used as-is.

Due to temporal correlation training and validation cannot be divided over random chunks or batches, according to the 'traditional' 30%–70% chunks (Gholamy et al., 2018). Therefore, the training data is split into equal years instead, as shown in Figure 6. Data outside the training period is used for validation. This includes the period before the training period, when available.

The robustness of the model to the selection of the training data is assessed from the stability of the results when training 260 over the subsequent periods (Figure 6), a variation on cross-validation (Krkač et al., 2020). Each model iteration starts with the same (random) initial weights, but is trained independently from the start. The quality of fit is assessed by evaluation of the loss function, the mean squared error, on the periods not used for training. Finally, the model performance is compared between the training periods. Large deviations of the model quality suggest there are dynamics the system is not capable to describe.

To assess the impact of irrelevant data on the system, as well as the effect of over-fitting, the additional, correlated random 265 variable (`fake`/`fake`) is used. Over-fitting will make the model prone to spurious correlation with this variable, that results in poor performance in the validation stage. Furthermore, to ensure there is no accidental correlation between the seasonal noise and the deformation signal during training and/or validation, the signal was re-rendered for every model run.

All possible combinations of the eleven input variables were tested on the model. With eleven variables this results in $2^{11} - 1 =$ 2047 combinations, as each of the time series may be used or not (2 options), expect for the case where no input is used. 270 Furthermore, the model was trained and validated on each of the ten combinations of training and validation year(s). Each



sequence of model training and validation was repeated at least three times, to account for the 'luck' introduced by the random initialization of each model. In total 147 984 model runs were performed.

## 5   Results

The best solution out of all model runs, judged on the minimal mean squared error on validation, is based on a single LSTM-
node and only four of the eleven input variables available: precipitation from GPM (V2); soil moisture from SMAP (V5) and ERA5 (V7); and evaporation from GLEAM (V8), where the numbers refer to Table 1. The minimal mean squared error on validation was achieved when the model was trained over period 3 (Figure 6, 2016-05-04–2019-05-04), the mean squared error of this model run was $1.03 \frac{\text{cm}^2}{\text{year}^2}$, below the average of $3.15 \frac{\text{cm}^2}{\text{year}^2}$ ($\sigma \approx 1.3 \frac{\text{cm}^2}{\text{year}^2}$, from 1718 samples) for this model configuration.

The full nowcast is shown in Figure 7, including the training period shaded in gray. Although, based on visual inspection, reasonable results are achieved in summer and autumn, the nowcasting model is unable to predict the deformation rate in winter and spring in the training period. Especially surprising is the jump in the winter of 2018/2019, where a strong acceleration is predicted which does not occur until early summer. The validation period, from 2019-05 onwards, shows little variation. The deceleration in the summer and autumn of 2019 is overestimated and shifted, likewise the acceleration in the December 2019
is predicted correctly, but too early. Overall the predictions show long-term stability (Figure 8) as enforced by the choice of the mean squared error as loss function.

The modelling results are overall unsatisfactory: the acceleration and deceleration are typically not predicted timely, or not at all. This is surprising in the light of the success reported by others (Table C.1). Although we designed our model to match our understanding of the interplay of hydro-meteorological conditions and deformation, the physics behind slope processes at the
Vögelsberg, the model was unable to capture this relation. The deformation at the Vögelsberg is driven by a complex interplay of hydro-meteorological conditions, unlike most of the examples in Table C.1, that often includes a strong, stable driver, such as a reservoir. This lack of such a single, strong, driver complicates the working of our data-driven model.

### 5.1   Contribution of individual variables

Due to the complexity of the operations applied to the input signal in the LSTM layer, it is not straightforward to analyze the
contribution of the individual components to the final model outcome. As all model variations were tested (§4.4), it is possible to analyze the influence of the presence of a variable by comparing the quality of the model variations. For this analysis only model iterations with a training period (Figure 6) that left at least one year left for validation were used. Furthermore, all model variations were run multiple times to assess the robustness of the outcome to the random initialisation.

Figure 9 shows the results of this analysis, and illustrates the mean squared error over the validation period for all models
including each variable. For each variable the minimum and average mean squared error for the validation period are shown,





**Figure 7.** Result of the deformation nowcast, run of the full time frame of the available deformation time series. The shaded time span was used for training. Shown as thin lines are the subsequent, daily, nowcasts for benchmarks 'D5_1' and 'D_WS_1'. Per day four deformation nowcasts are shown, with the start of each line being the day after the day the nowcast was issued. Note the warm-up time at the start, without predictions, that is required to fill the memory of the LSTM-node. The final nowcast ends four days after the end of the reference measurements.





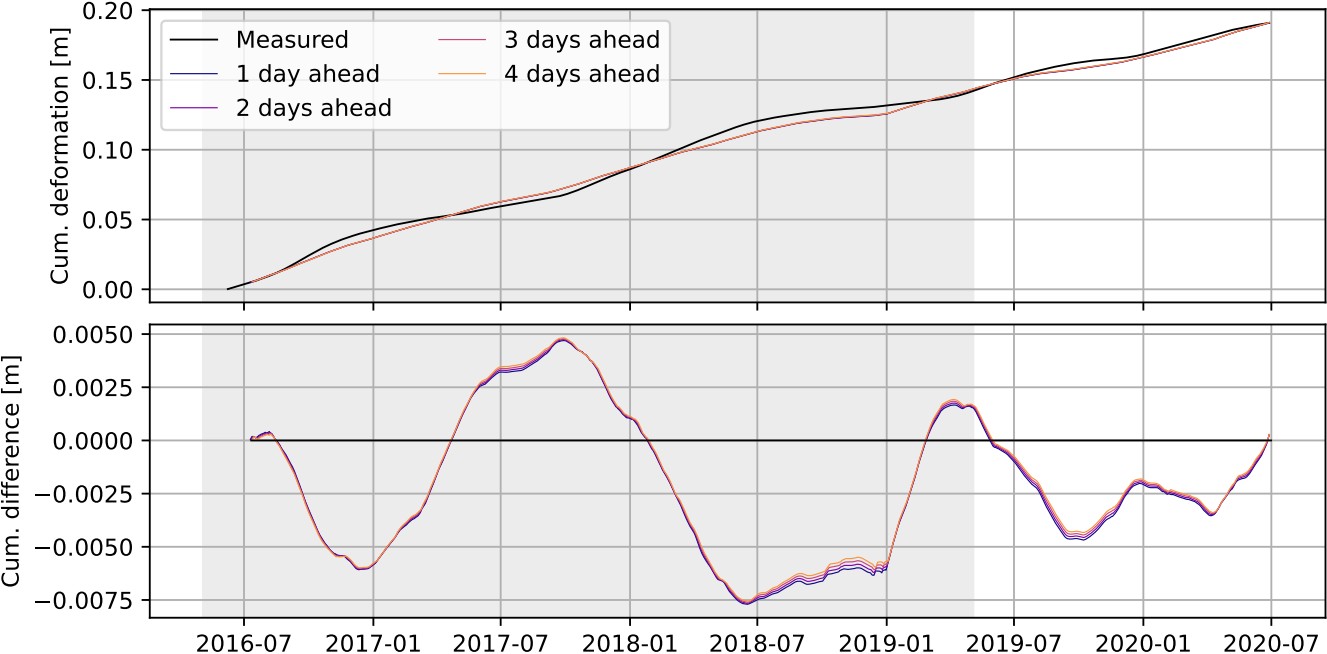

**Figure 8.** The cumulative deformation, as predicted by the consecutive, individual model runs closely matches the observed deformation over the full four years of deformation measurements. The difference is calculated as 'modelled - observed' ($\hat{\mathbf{y}} - \mathbf{y}$) cumulative deformation.

while the maximum mean squared error is often out of range. The thickness of the line indicates the density of results for that mean squared error, where thicker lines at lower mean squared error indicate a concentration of models with high quality of fit.

Models based only on `SMAP/sm_profile` (V5) score the poorest (highest mean squared error) on average, but with the widest distribution, including many solutions with a low mean squared error. The difference in performance between the
variables vanishes as more variables are introduced into the model, however, the models including the SMAP soil moisture (V5) time series show a consistently larger range in performance, including models with a low mean squared error. Remarkable is the approximately equal performance between `API/API` (V10) and `fake/fake` (V11), where the latter contains no information on the hydro-meteorological processes and is only marginally outperformed by the Antecedent Precipitation Index (API, V10). For models with more than four variables, there is no significant difference in model quality for any of the variables.

**6   Discussion**

We believe the unsatisfactory performance of the model has three root causes: i) the inability of the model to capture the complex dynamics of the system; ii) the limited quantity of training data available to this type of problem; and iii) the limited, noisy representation of the slope dynamics in the available remote sensing data. Most natural deep-seated landslides are characterized



**Figure 9.** Violin plots of the mean squared error for model variations with one to four variables, including the variable listed. For more than four variables the relative importance of the individual variables to the model quality becomes insignificant.





**Table 2.** List of reference models tested for comparison to `lstm1-32`. Their performance is shown in Figure 10. To calculate the number of model parameters: $n$ the length of the time series provided to the model, $k$ the number of input variables, $m$ the number of memory cells, and $h$ the number of hidden nodes. A single hidden layer is assumed. The number of parameters includes the final, output layer of four nodes for each of the two deformation time series.

| Model | Hidden layer | Activation | History | Parameters |
|---|---|---|---|---|
| `lstm1-32` | LSTM (1 memory cell) | tanh | 32 days | $4(k+m+1)h + 2(4 \cdot m + 4)$ |
| `lstm3-32` | LSTM (3 memory cells) | tanh | 32 days | "           " |
| `rnn1-32` | RNN (1 memory cell) | tanh | 32 days | $h(k+m+1) + 2(4 \cdot m + 4)$ |
| `rnn3-32` | RNN (3 memory cells) | tanh | 32 days | "           " |
| `rnn3lin-32` | RNN (1 memory cell) | none | 32 days | "           " |
| `da-32` | 8 cells | none | 32 days | $h \cdot k + 2(4 \cdot h + 4)$ |
| Lin. Least Sq. | none | none | | $2(n \cdot k \cdot 4 + 4)$ |

by a complex interplay of causal (antecedent) and triggering conditions: this is also true for the Vögelsberg landslide. However,

we believe that it is exactly these challenges that we should aim to tackle with a machine learning model approach.

## 6.1   Model configuration

The possibilities for data-driven modelling are infinite: our model is only a single realisation of the possible combinations of variables and operations. This raises three questions regarding the model selection: i) how to match model and process; ii) how to validate and quantify the quality the nowcast; and iii) how to tune the model implementation.

Major challenge for the model of a deep-seated landslide is the discrepancy between the sub-daily variations of the input (especially precipitation and snowmelt), and a delayed, daily output (accelerated deformation). Therefore, non time-aware models show erratic behaviour, as the consequence of sudden changes to conditions such as snow cover and as well as (extreme) precipitation, that, in reality, do not translate into immediate acceleration. Traditionally, the addition of groundwater physics, smoothing the hydro-meteorological signal, circumvents these peaks. However, the addition of groundwater physics requires

knowledge of the geohydrology of the specific slope.

An LSTM-node resembles a bucket model, and was chosen such to capture the delay between precipitation and deformation, by modelling the build-up of water in the model. Our results showed that our model was unable to fully capture these hydro-meteorological dynamics. For reference five alternative models were implemented (Table 2), that were designed to better address the diversity of the slope, and/or lower the number of parameters required by the model.

The `lstm3-32` model contains two additional memory cells (buckets) in the LSTM-node, compared to the `lstm1-32` model previously used. The concept is that the memory cells may represent different systems or layers in the subsurface, potentially interacting with each other. For each subsequent time step, all states are included in the calculation of the new states, and



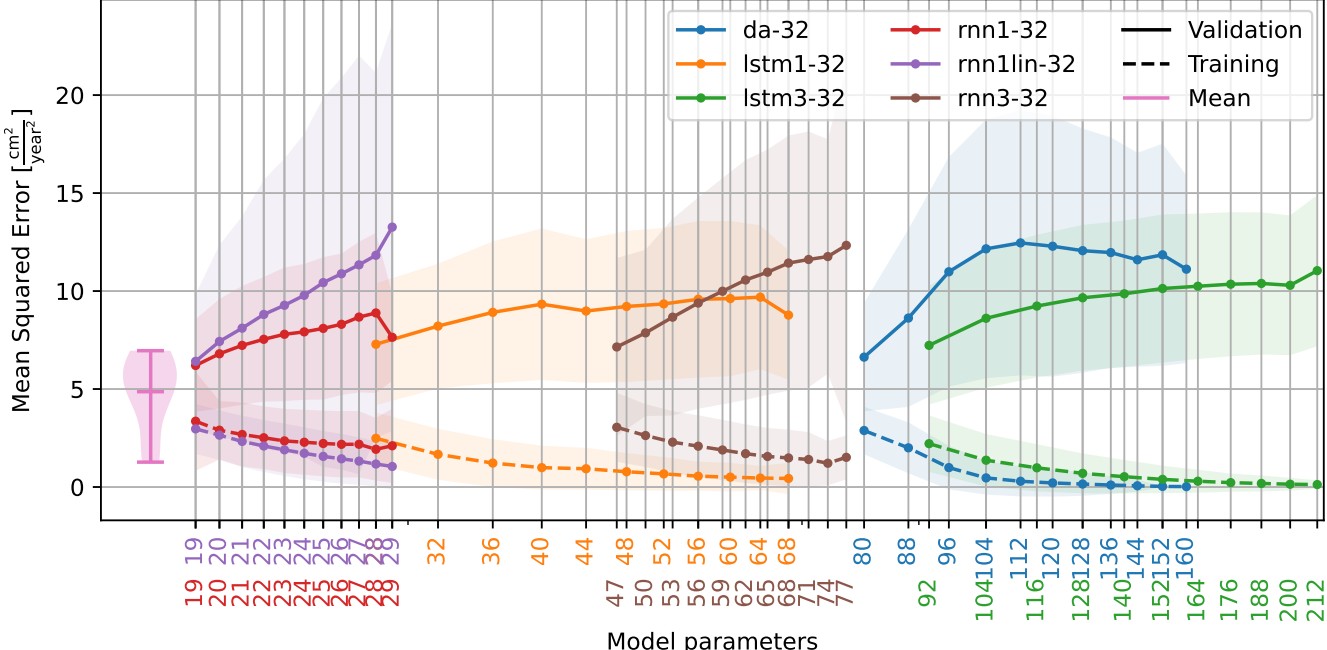

**Figure 10.** Relationship between the number of model parameters and the quality (mean squared error) of training and validation as extracted from the 147 984 model runs. The number of parameters is related to the number of input variables. For LSTM based networks, for example, there are four parameters per input variable per LSTM memory cell required. Note the logarithmic scale on the x-axis. See also Figure 13, on the relationship to the number of parameters. On the left, the mean squared error is shown would the mean deformation rate be used to nowcast, based on all nine training periods (Figure 6).

could therefore also model interactions between layers in hydrology, such as the transfer of between layers. The `rnn1-32` and `rnn3-32` models based on a traditional Recurrent Neural Network are similar to their LSTM counterparts, with one and three memory cells respectively. However, unlike an LSTM-node, they are unable to 'forget' their state on command, and are more susceptible to unstable behaviour. The `rnn1lin-32` did not incorporate an activation function and is comparable to a moving average filter with interaction between the variables. For all three models the number of parameters is less than for the equivalent LSTM based models.

The `da-32` model resembles a linear least squares model. Variables are first summarised as their average over their 32-day history, and included in eight nodes without bias in the hidden layer of the network. The final predictions are a linear combination of the node values. In a 'traditional' linear least squares solution, a direct combination of all input variables, the number of parameters will often outnumber the number of observations available, and was therefore not tested.

The performance of each model is shown for comparison in Figure 10, as function of the parameters required. Model performance is typically optimal for models with only a single parameter, and is comparable between the models. Like the original model (`lstm1-32`), each model was re-run multiple times with a random initialisation of the seasonal noise (V11) and model








parameters, to verify the consistency of the output. Most alternative models do not outperform the average deformation rate as predictor for the future deformation rate, as shown in Figure 10.

### 6.1.1 Performance metric

For early warning systems, prediction of the onset of acceleration (Figure 11) is more important than the deformation quantity.

However, false alarms, triggered by insignificant accelerations, may undermine confidence in the early warning system. At this stage of development, we would rely on professional interpretation by an expert to limit the number of false alarms. However, the system should warn the expert for potentially bad predictions, for example due to previously not encountered conditions. The timing of the nowcast should allow for further analysis of the prediction without jeopardising precautionary measures for accelerated deformation.

This leads to five desired properties for the nowcasting system: the system should i) predict onset of acceleration; as well as ii) the maximum deformation velocity; iii) four or more days ahead that deformation will begin; iv) predict when the slope is 'stable' again; and v) quantify the certainty in the prediction. Unlike most estimation problems, the timing and not only the quantity of the predicted deformation is important to the user. An acceleration phase predicted too early or slightly late may still trigger the desired alertness, and still serve a purpose, even though the predicted amplitude on that day is wrong.

Performance indicators typically used in landslide studies, such as the ROC curve (Corsini and Mulas, 2017), are not applicable to deformation velocity estimates as our nowcast is not a binary classifier, nor is there a single deformation event. Studies on deformation nowcasting typically rely on the correlation coefficient ($R^2$), that serves as an indication of how well the nowcast predicts the excursions of the deformation rate from the mean. For the nowcast presented, the correlation coefficient $R^2 \approx 0.31$, and is approximately equal for all four days. The correlation coefficient, however, neglects the amplitude and the non-zero 365 average deformation of the slope.

A 'standard' error metric, e.g. the mean squared error, is sensitive to the mean as local optimum, but is unbiased and therefore stable in the long term. As an alternative such error metric could be evaluated at 'peaks & valleys', the peaks of the deformation rate, only, emphasizing extremes and disregarding their onset. With this method there are less samples, only the extremes, but they are less correlated and include the amplitude of the event. Although this captures the timeliness of the extremes, it 370 disregards timing of the onset and pattern of the acceleration phase. Moreover, this approach requires information on the peaks and valleys, and that those are correctly identified beforehand.

Due to the lack of information on the extremes of the deformation, we chose to use the mean squared error as error metric. This metric ensured a long term stability, and connected stability of the deformation nowcast, as demonstrated by the cumulative deformation (Figure 8). As a consequence, the system preferred 'average' solutions, overestimating the deformation rate in 375 stable periods and underestimating the deformation rate in periods of accelerated deformation.



### 6.1.2 Variable pre-processing

All time series of the input data, compare Table 1, were offset to be zero-mean and scaled such that the standard deviation was equal to 1. As linear scaling of the variables is applied within the LSTM-nodes, scaling of the input variables should not be necessary. However, the scaling creates a level playing-field for the random variations in the initialisation and therefore may

benefit network training. Moreover, by offsetting the data to the average, the values are represented as deviations from their average condition, in the absence of a per variable offset in the network's LSTM-nodes.

Additional variables may be derived from the direct observations. In our model, the Antecedent Precipitation Index (API) is such derived observation, and was chosen to enhance the information content of the hydro-meteorological observations to the model (i.e. provide higher predictive power to the model). This 'feature generation' is an important component of more

traditional machine learning techniques, where the system is not expected to derive those relations autonomously. Derived, additional features were extensively used by Krkač et al. (2020, 2017), for example, who created additional features to capture the conditions on the landslide, or Miao et al. (2022) who derived ten features from only two sources (rainfall, reservoir level). Drawback of the addition of large quantities of such derived variables to the system is that each additional time series requires additional model parameters to be optimized.

### 390 6.1.3 Handling unencountered conditions

Given the limited availability of deformation measurements, most of the data is required to train the model. Moreover, the variation in conditions is limited to the variation in those five years. It is therefore likely that the model will encounter conditions in operation that it had not encountered before. The continuous nature of the model proposed, and the alternatives discussed in §6.1, the output for such conditions is not bound to the previously encountered conditions.

For simple combinations of variables, i.e. of a single or a few variables, the response may be tested empirically. Note that the full 32-day history has to be included in this simulation. However, the response may not be so straightforward: a warm summer day combined with hail from a thunderstorm may trigger an unrealistic 'path' in the model. Therefore, for more variables, the number of potential combinations increases drastically and may no longer be feasible to simulate.

Predictions of extraordinary responses are not necessarily undesirable, an unbound acceleration, i.e. landslide collapse, predic-
tion should be possible. However, the model would preferably warn for a potential unstable state of the nowcasting system. This could be achieved by an ensemble of models, either based on the same model, or model variations. Especially models with different time series lengths may be able to help pinpoint the source of the discrepancy.

### 6.1.4 Spatial distribution

Our model of the Vögelsberg is based on two benchmarks, that are on two distinct sections of the slope (Figure 1) that have
shown to exhibit different deformation behaviour. The southern, inhabited, part of the slope exhibits constant deformation, with limited acceleration in wet periods. In contrast, the benchmark on the northern part of the slope, shows strong accelerations




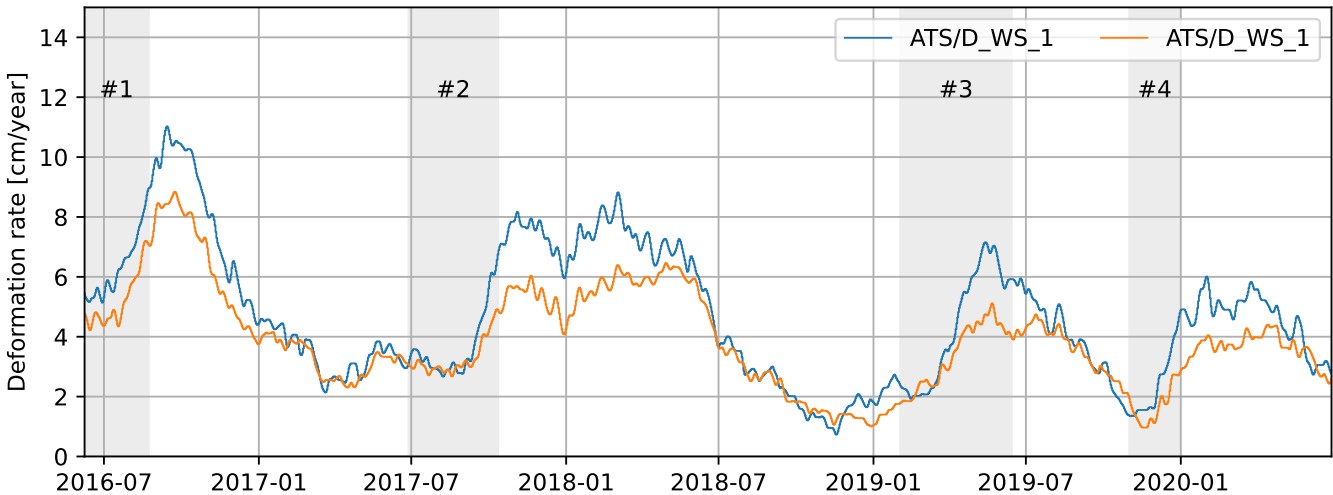

**Figure 11.** Three acceleration events (#1, #2 & #3) at the Vögelsberg, as identified by Pfeiffer et al. (2021). The fourth acceleration period (#4) was identified in the data acquired after Pfeiffer et al. (2021).

and deceleration as a delayed response to strong precipitation (Pfeiffer et al., 2021). The location of the benchmarks on the slope is not provided to the model, and the two benchmarks are treated as two parallel results of the same LSTM-node.

As an alternative, a location index could be specified, for example as binary indicator of the landslide section, or as continuous
signals such as a distance to the centre. Instead of two or more predefined outputs from the same model, a single model may handle different benchmarks differentiated by additional input variables encoding their position within the system. However, given the shallow model design, care should be taken to design the model such that this index works as a scaled multiplier of the hydro-meteorological conditions.

**6.2  Limited number of distinct events**

Over the full time span of the measurements, four distinct acceleration periods can be identified (Figure 11). Especially these acceleration periods are of interest to an Early Warning System, as they mark the start of a period of accelerated deformation and associated hazard. Although the periods of accelerated deformation are comparable in length to the periods of continued, but reduced, deformation, the acceleration events are much shorter (Figure 11). Therefore, these periods are underrepresented in error metric during training and validation. However, training on these four periods alone leave insufficient variability to
describe the system and reliably fit the required model parameters. Furthermore, the episodic deformation behaviour poses a challenge to the prediction system since the forcing variables on the slope do not reflect such sudden changes observed in the deformation behaviour, as shown in Figure 3.





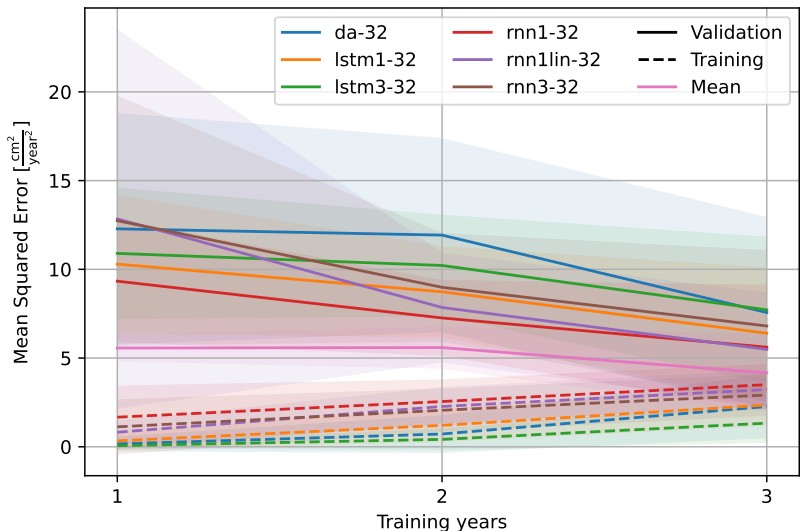

**Figure 12.** Length of the training region, aggregated to (approximate) years, compared to the quality of fit of the model, measured as the mean squared error. An increase in model fit is visible with the increase in training length, however, most models are outperformed by the mean deformation rate as a predictor.

### 6.2.1 Length of training

The deformation data and nowcast results are strongly correlated: a forecast three days ahead describes the same day as a
prediction two days ahead the next day. Given there is more than a single degree of freedom in the model, without prior knowledge of the process, there is no predictive power in a single acceleration event. Hence, multiple events are required to properly train complex models, in the absence of constraints on the process/model. As a consequence, due to the limited variety of events in the training data, the predictive power of the nowcasting system may be reduced, due to over-fitting on the characteristics of these events only.

To test the effect of the training length on the models, the models were trained on nine of the ten training periods identified in Figure 6 that had a least a year left for validation. The mean squared error, measured on the training as well as validation period, is shown in Figure 12. The results are consistent between the models: all models show that as the training period increases, the quality over the training period decreases (dashed line, increasing mean squared error) due to the increased variability of the events therein. Likewise, the quality over the validation period increases (solid line, decreasing mean squared error), as
the model generalizes better. This is also reflected in the lower standard deviation for validation over longer training periods. Hence, a longer training period makes the system more robust against the variations encountered by the system.

To train and validate the nowcasting system, the time series was subdivided in calendar years measured from the start of the measurements. An alternative, common subdivision would be in hydrological years or water years, that are typically defined to be from October 1 to September 30 and divided by the precipitation minimum (Lins, 2012). This subdivision is typically





applied to cut the data in a hydro-meteorologically quiet period of the year. However, the strong deformation events in period 1 and 2 overlap with this subdivision. Furthermore, with this subdivision, only three periods would be available, instead of four. Moreover, Parajka et al. (2009) show that the period of minimum precipitation cannot be pinpointed to a single winter month. Therefore, the decision was made to align the training years with the measurements instead.

### 6.2.2   Noise reduction of the deformation signal

Essential to the success of the nowcast are the properties of the signal to be predicted. The effect of noise in the deformation signal on the modelling is twofold: first, random perturbation complicates the training by masking the best solution, and, second, leads to an underestimation of the final quality of the model during validation. Hence, the noise in the deformation signal defines the upper limit for the quality of the deformation estimate. Up-slope deformation, present in the raw deformation time series, was considered to be unrealistic and therefore noise by definition. Under the assumption that the noise is unbiased,

the noise will be reduced in averaged samples. Therefore, a moving average filter was applied to the deformation time series with increasing length until no negative deformation remained.

The model was developed with the requirements for an operational system in mind, restricting the system to only use historic observations at any point in the process. Inclusion of future samples would require the system to react to future conditions that have not (yet) been observed on the slope: any filtering, such as smoothing, should not drag future observations back in time.

Therefore, the moving average filter cannot be centered, and averaging is applied to the preceding 31-days, rather than $\pm 15$ days around the current time step as would be possible in re-analysis.

The variation in the deformation signal at the Vögelsberg is relatively small, in deviation from a long term trend. Due to the milimeter-scale measurement uncertainty in the deformation measurements, the deformation signal is dominated by noise on the short time scale of days to weeks, the relevance of a deformation prediction on a daily basis is doubtful. Furthermore, due

to the inertia of the landslide body, as well as smoothing of the deformation measurements, accelerations and decelerations are spread over adjacent days (Figure 6) and the amplitude of the acceleration is lost. For a successful, daily application, a clear separation between events and noise is required (higher SNR), either due to a faster process, or due to reduced noise in the deformation observations.

### 6.3   Input variables

The variable selection in Table 1 was compiled based on our knowledge of the physics behind the landslide process, as well as the availability and continuity of the data. With the ambition for a future, regional implementation in mind, the variables preferably come from satellite remote sensing observations rather than local, field sensors. However, we did not succeed in a fully remote sensing driven operation, due to the limited availability of such operational products. Especially deformation observations from space ('InSAR') were found to be promising but we were unable to replace our local deformation time series

with the noisier satellite deformation data.





### 6.3.1 Availability of variables

The model was designed under the assumption that data from all sources is continuous and readily available to the system. Traditionally, local weather and groundwater monitoring stations provide timely, local, high quality observations. However, such monitoring stations are not available everywhere. Out of the variable selection (Table 1) only GPM (V2) and SMAP (V5)
satisfy this condition and provide operational data products, that could be integrated in a nowcasting solution.

For a successful integration of satellites observations in an operational nowcasting system, a high, sub-weekly, update frequency is required. However, most remote sensing products were available at a delay of days to weeks, still too late for integration in a nowcasting system. As a consequence, the variable selection in Table 1 contains variables that are only available in yearly iterations (e.g. GLEAM).

Satellite radar interferometry (InSAR) is a proven method for landslide deformation monitoring (Colesanti and Wasowski, 2006; Hilley et al., 2004). However, especially mountainous environments create a complex interplay of local atmospheric effects and topography (Hanssen, 2001). A feasibility study showed that the slope orientation and topography would allow for the application of satellite radar deformation measurements at the Vögelsberg (van Natijne et al., 2022). Further processing demonstrated the presence of persistent scatterers on and around the houses at the slope, the objects of primary interest.
However, the use of satellite based InSAR as source of the deformation measurements was not feasible, due to the low temporal resolution, as well as the noise in the deformation signal (Zieher et al., 2021).

### 6.3.2 Data continuity

Temporal continuity of input data is required to provide the model with consistent samples of the slope conditions. Short periods of missing data, e.g. days, may be forward filled, but will reduce the data quality for the full integration length (i.e.
32 days). Observations received late may still be updated in later iterations, to mitigate this effect. However, what to do with missing data: a single day or a whole season, or the termination of a data source, for example due to satellite failure? As a fallback one could model and train systems with different variable combinations in advance, and nowcast based on the best model available for the variable combination available in the 32 days prior.

The LSTM-nodes may be implemented in a stateful fashion, where the state of the hidden nodes is retained after each predic-
tion. Such implementation is more computationally efficient, as each subsequent nowcast will require only a single pass over the most recent data. In such implementation, however, discontinuous or erroneous variables may have a lasting effect on the model memory. Therefore, the system was based on continuous re-initialisation with a 32-day observation history instead. The computational drawback is limited, given the small scale of the model, and is acceptable in the light of the greater operational flexibility.





### 6.3.3 Variables not related to the hydro-meteorological cycle

Indirect observations of the hydro-meteorological cycle may still prove valuable to the nowcasting system. The temperature, for example, may serve as a proxy indicator for evaporation. Temperature is related to the seasons in most climates, and therefore there will be a correlation with the season (day-of-year) as well. However, extra care should been taken including variables that describe the typical/average condition, such as the season. Such variables do not capture the current dynamics of the system and may only describe average conditions, and constrain the system in extraordinary circumstances. The Vögelsberg landslide is known to be sensitive due to changes in the ground water level, irrespective of the season.

### 6.3.4 Input variable selection

The success of a data-driven model lies in the (expert) selection of the input data. Unrelated variables make the system prone to spurious correlations, especially with limited training data compared to the degrees of freedom in the model or if the method is unable to discard or otherwise ignore sources with low information content. Furthermore, unrelated input variables, or even just noise, should not yield sensible results: "garbage in, garbage out".

The effect of noise in the conditions was tested by the inclusion of a Brownian motion signal (see §4.4), that does not have a relation to the system, except for basic properties (i.e. mean, standard deviation, autocorrelation period) similar to the input variables. Any model run including this signal should not outperform an otherwise comparable model without this variable. However, many of the models did, especially when many ($\geq 5$) variables were included, where it helped to create unique variable combinations and allowed the model to over-fit.

Parameters on geology and topography were left out of the selection, and assumed static. However, neither were land cover changes included. In the case of the Vögelsberg, it was known that little changes were to be expected over the time frame of the measurements available. An alternative to the inclusion of such variables is to frequently re-train the model on a recent section of the time series only to adapt to changes. However, although the system will adapt to changing dynamics, re-learning will mask the drivers behind long term effects, and/or adapt too swiftly, for example to seasonal differences, reducing the overall model quality. Land cover changes will not be uniform across slopes, and act on different time scales (e.g. neglected pasture fields versus forest fires). Moreover, especially in regional studies, the land cover and land cover change may not be comparable between slopes.

To limit the number of variables, only the observation or modelling result closest to the Vögelsberg was used from regional products. However, as Pfeiffer et al. (2021) found, precipitation and snow-melt higher up in the catchment is relevant for the system (Figure 1). Based on the typically low ($\simeq 10$ km) spatial resolution of the variables (Table 1), we deemed this assumption justified for this selection of variables. When higher resolution observations are added, this should be reconsidered, and additional points may be added as extra variables.





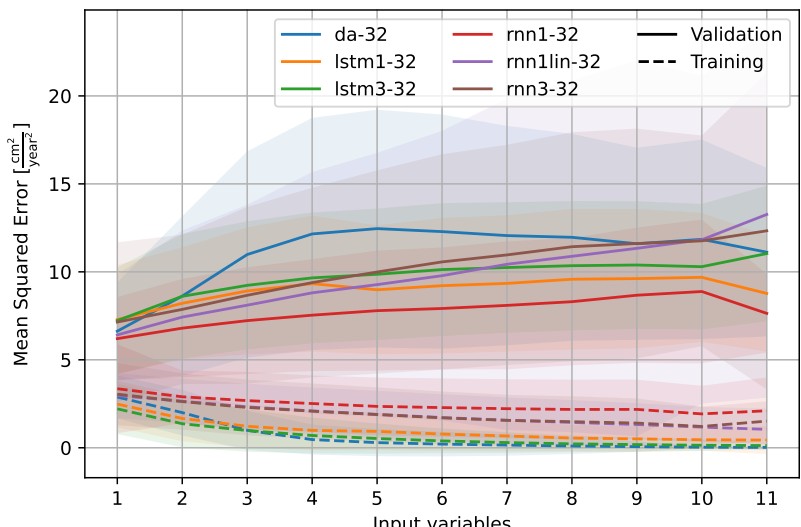

**Figure 13.** Relationship between the number of input variables and the quality (mean squared error) of the training and validation as extracted from the 147 984 model runs. The mean squared error on validation increases slightly as more input variables and associated model parameters are introduced, likely due to over-fitting of the model on the training period facilitated by the extra complexity of the model.

## 6.4 Outlook

Our results show that deformation nowcasting is an open challenge. Although well monitored, the Vögelsberg landslide is a complex system, and therefore not a straightforward test case. Our results are inconclusive whether our method could work on other deep-seated landslides. More direct dynamics, and/or stronger and more frequent acceleration periods would help constrain the system. The inclusion of field data, such as groundwater level (Krkač et al., 2020), might be another approach to bypass modelling of the most volatile hydrological processes. The ideal slope to further develop a machine learning based nowcasting method has the following characteristics: i) a dynamic deformation behaviour; ii) is controlled by hydro-meteorological conditions, with limited delay; and iii) has field monitoring data for reference and training.

For short time series machine learning methods are known to be outperformed by basic statistical methods (Makridakis et al., 2018). Therefore, our current challenge to nowcast deformation time series may be partially solved in the near future by the natural extension of time series. Furthermore, continued development of the (satellite) data products by their providers may enable new possibilities. Desirable improvements include timeliness of delivery of data products, as well as their precision and spatio-temporal resolution.

Notable is the recent publication of the first version of the European Ground Motion Service data set (Crosetto et al., 2020), a pan-European InSAR product. This data set will allow for experimental, regional, weekly nowcasting systems based on a replay of historic observations. Regional applications will enhance training possibilities and may help overcome the hurdle





of limited deformation time series, as multiple slopes are monitored simultaneously. However, to 'learn' from the differences between slopes, and enlarge variation in training data, events have to be largely uncorrelated.

## 7  Conclusions

Although the Vögelsberg is a well monitored landslide, the number of recorded acceleration events, within the available four
years of daily deformation measurements, is limited compared to other machine learning problems. A simple, time series capable, model with limited parameters was required, therefore, we designed an LSTM-based machine learning algorithm to nowcast the deformation of the Vögelsberg deep-seated landslide from the conditions on the slope. The algorithm was trained on maximum three years of deformation observations and satellite observations of relevant hydro-meteorological conditions at the slope. The best model configuration and variable combination was determined by cross-validation with 147 984 model
variations.

Although rooted in the landslide dynamics, even our best model was incapable of capturing the versatility of responses on the Vögelsberg, and convincingly predict the landslide deformation rate at the Vögelsberg four days ahead. Especially the four acceleration events were not predicted timely, although the overall amplitude of the prediction was successfully enforced by the mean squared error as loss function. The Vögelsberg landslide showed versatile dynamics, where the full range of
slope dynamics and responses to the hydro-meteorological conditions were not present in the available data. Therefore, the slope processes were too complex to model the landslide deformation from satellite surface observations, given the limited observations of acceleration events. Hence, the machine learning model was incapable of 'understanding' the relation between conditions and deformation.

Deformation nowcasting will be a necessity for regional or even continental landslide monitoring and early warning systems.
Satellite remote sensing has the potential to provide longer time series, over wide areas. This leads us to the general recommendation for the application of machine learning to reactivating, deep-seated, landslides: improve data quality, and lengthen the deformation time series. The ideal landslide for further development of deformation nowcasting: is highly dynamic (many events to train on), has a limited delay between forcing conditions and deformation, is well monitored, and is not catastrophic.





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

## Appendix A: Data

See Figure A.1.

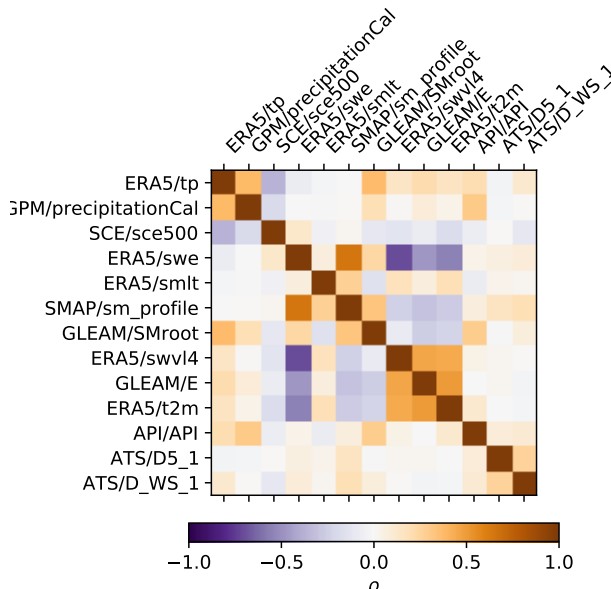

**Figure A.1.** Correlation between variables.

## Appendix B: Total Station

See Figure B.1.





**Figure B.1.** Smoothed deformation signal, shown for an increasing length (in days) of the moving average filter. The filter only includes historic observations, and is not 'centred', to match the properties of an operational system. The increasing time lag is visible for the subsequent filter lengths by the right shifting of the velocity peaks. For initial observations, a filter length of half the final length of the filter was accepted.



## Appendix C: Models

### C1 State-of-the-art

See Table C.1.

*Author contributions.* **A.L. van Natijne**: Methodology; Software; Formal analysis; Writing - Original Draft. **T.A. Bogaard**: Conceptualization; Writing - Review & Editing; Supervision. **T. Zieher**: Resources; Writing - Review & Editing. **J. Pfeiffer**: Resources; Writing - Review & Editing. **R.C. Lindenbergh**: Writing - Review & Editing; Supervision.

*Competing interests.* The authors declare that they have no known competing financial interests or personal relationships that could have appeared to influence the work reported in this paper.

*Acknowledgements.* This work was carried out under the framework of the operandum (OPEn-air laboRAtories for Nature baseD solUtions to Manage hydro-meteo risks) project, which is funded by European Union's Horizon 2020 Framework Programme for research and innovation under grant agreement 776848.

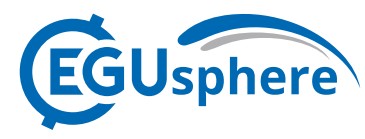

**Table C.1.** Examples of different integration methods, linking hydro-meteorological conditions to deformation time series, and associated case studies. Where applicable the reference methods used in the paper are listed in brackets. Updated after van Natijne et al. (2020).

| Reference | Case study | Country | Obs. driving forces | Deform. meas. | Method (reference methods) |
|---|---|---|---|---|---|
| Miao et al. (2022) | Baishuihe, | China | Rainfall, reservoir level | GNSS | FOA-BPNN + polynomial trend |
| Zhang et al. (2021) | Fengning, | China | Rainfall, toe excavation (incl. blasting) | Total station, inclinometer, fissure meter | LSTM, BP, SVM, ARMA |
| Deng et al. (2021) | Hollin Hill, | United Kingdom | Rainfall, acoustic | Inclinometer | ELM, NN, SVM, LASSO-ELM |
| Li et al. (2021) | Baishuihe & Bazimen, | China | Rainfall, reservoir level | GNSS | GA-CEEMD-RF |
| Liu et al. (2020) | Baishuihe, Bazimen, Baijibao, China | China | Rainfall, reservoir level | GNSS | LSTM, GRU, RF + polynomial trend |
| Krkač et al. (2020) | Kostanjek, | Croatia | Rainfall, groundwater | GNSS | Linear regression, RF |
| Bossi and Marcato (2019) | Passo della Morte, | Italy | Rainfall, groundwater | Inclinometer | Linear regression |
| Li et al. (2019) | Baishuihe, | China | Rainfall, reservoir level | GNSS | DBN + trend (wavelets) |
|  | Longnan, | China | Rainfall | InSAR | Linear trend + wavelets |
| Liu et al. (2021)[a] | Tanjiahe, | China | Rainfall, reservoir level | GNSS | LS-SVM, ELM |
| Wang et al. (2019) | Laowuji, | China | Rainfall, toe excavation | Total Station | LSTM |
| Xie et al. (2019) | Baishuihe & Bazimen, | China | Rainfall, reservoir level | GNSS | LSTM |
| Yang et al. (2019) | Majiagou, | China | Rainfall, reservoir level | Inclinometer | LSTM |
| Zhang et al. (2019) | Baishuihe, | China | Rainfall, reservoir level | GNSS | Correlated grey model |
| Li et al. (2018) | Baishuihe, | China | Rainfall, reservoir level | GNSS | LASSO-ELM, Copula (ELM, SVM, RF, kNN) |
| Miao et al. (2018) | Baishuihe, | China | Rainfall, reservoir level | GNSS, inclinometer | GA-SVR, GS-SVR, PSO-SVR |
| Huang et al. (2017) | Baishuihe & Bazimen, | China | Deformation | GNSS | ELM, PSO-SVM |
| Krkač et al. (2017) | Kostanjek, | Croatia | Groundwater (change), season | GNSS | RF |
| Logar et al. (2017) | Ventor, | United Kingdom | Rainfall | Crackmeter | ANN |
| Ma et al. (2017) | Zhujiadan, | China | Rainfall, reservoir level | GNSS | Decision tree |
| Wen et al. (2017) | Shuping, | China | Rainfall, reservoir level | GNSS | GA-LSSVM + polynomial trend |
| Zhu et al. (2017) | Kualiangzi, | China | Rainfall | GNSS | DES-LSSVM, GA-LSSVM |
| Cai et al. (2016) | Xiluodo, | China | Rainfall | Extensometers | GA-LSSVM |
| Cao et al. (2016) | Baijiabao, | China | Rainfall, groundwater, reservoir level | GNSS | ELM (SVM) |
| Zhou et al. (2016) | Bazimen, | China | Rainfall, reservoir level | GNSS | PSO-SVM (GA-SVM, GS-SVM, BPNN) |
| Jiang and Chen (2016) | Baishuihe & Liangshuijing, China | China | Rainfall, reservoir level | GNSS | GRNNS |
| Lian et al. (2015) | Baishuihe & Bazimen, | China | Rainfall, reservoir level | GNSS | LSSVM, ELM, combination |
| Ren et al. (2015) | Shuping, | China | Rainfall, reservoir level | GNSS | PSO-SVM + logistic trend |
| Liu et al. (2014) | Baishuihe, | China | Deformation | GNSS | SVM, RVM, GP |
| " | Super-Sauze, | France | Deformation | Extensometer | SVM, RVM, GP, polynomial |
| Chen and Zeng (2013) | Baishuihe, | China | Deformation | Extensometer | BPNN |
| Du et al. (2013) | Baishuihe & Bazimen, | China | Rainfall, reservoir level | GNSS, inclinometer | BPNN |
| Lian et al. (2013) | Buishuihe, | China | Undisclosed? | | EEMD-ELM, M-EEMD-ELM (ANN, BPNN, RBFNN, SVR, ELM) |
| Corominas et al. (2005) | Vallcebre, | Spain | Groundwater | Extensometers | Physics |
| Neaupane and Achet (2004) | Okharpauwa, | Nepal | Rainfall, groundwater | Autoextensometer | BPNN |

[a] Analysis, allows for prediction.