# Peer review of "Machine learning based nowcasting of the Vögelsberg deep-seated landslide: why predicting slow deformation is not so easy"

_EGUsphere, 2022_

## Referee Comment (RC2)

Review of *Machine learning nowcasting of the Vogelsberg deep-seated landslide: why predicting slow deformation is not so easy*

The work presented by Natijne et al. provides an in-depth investigation into why their machine learning nowcasting model did not provide adequate results at the Volgelsberg landslide in Austria. The authors use a state-of-the-art machine learning model that incorporates detailed deformation data and remote sensed environmental data to try and predict deformation responses through time. The article is well-written and does a good job exploring the limitations of machine learning approaches. The article is important as machine learning methods are often heralded as a panacea. However, as this paper outlines, machine learning methods need to be applied to the right types of problems with the right type of data of adequate quality to provide effective solutions. While I believe the paper is well constructed and has valuable findings, I do have a few concerns that need to be addressed before publication. Below I provide a few general comments, followed by a line-by-line issues that should be addressed.

General comments:

1.  You provide a table of several research articles that have produced now-cast models. In section 2, you also say that at least some of these articles were for deep-seated landslides. I think it would be valuable for you to explicitly highlight what was different between this study and the ones that seemed to have success with now casting. Do you know exactly why they had success and you didn't? You discuss the reservoir being a factor for some of these studies, which makes sense. However, did any other studies try to nowcast deep-seated landslides with success?

2.  I'm a bit concerned by the environment predictor data you use in you model. The spatial resolution of the satellite data is often much coarser than the size of the landslide. I have a hard time seeing how these coarse resolution datasets could provide any meaningful information at the local scale you're looking at.

3.  I also have concerns with the deformation data itself. If the data is so noisy, are you sure you can trust it at all? Are you certain that the deformation signal you are trying to replicate isn't an artifact? If you are confident in the deformation signal, please better demonstrate why to the reader. For example, provide the accuracy of the data, the preprocessing step used by the Division of Geoinformation of Tirol, and elaborate on the corrections to the measurements. Based on the smoothing you conducted to get a usable signal, I find it difficult to trust this data. The poor quality deformation data may be partially responsible for your poor model fit.

4.  Other studies (e.g., Thomas et al., 2019; Yatheendradas et al., 2019) have assessed the utility of satellite-based weather data for slope stability and found mixed results. I believe its possible that many of your issues could be attributed the poor representation of satellite data for your study site. I think this merits discussion or justification for why you think this is not an issue.

5. I don't think section 2 in necessary for this paper. I found it to be a bit burdensome and ancillary to the main point of the paper. Consider trimming it down to only describing the points that are pertinent to the model you use and then putting it in the methods and/or discussion where you describe the model(s) you use in this study.

Line by lines:

11: Please clarify what you mean by "standard quality metrics".

35: "correlate *with* time-delayed".

50: Please provide citations for this claim.

58: Consider, "…may reveal slope processes *responsible for landsliding*"

68: Consider "…naive modelling. *That is, the model is* unaware…"

76: Please don't start a paragraph with "therefore". It is confusing as a reader. Reintroduce the idea you are discussing.

77: This is a sentence fragment. Please fix.

122: This sentence has a typo. I think you need to delete "in less".

123: Define epoch.

123: The number of data points required largely depends on the type of model used. Neural networks require a lot more than other methods (e.g., logistic regression), both of which fall under the term 'machine learning'. Consider being more specific by what type of machine learning method you're referring to.

Figure 1: Please put sub-figure labels on the figure and in the text to help orient the reader.

Put a box around the right sub-figure to show extent of the bottom left sub-figure.

On my computer, the colors of the land slide are not clear. I see red to the north, then yellow, then green going south-east. Also why are there three colors? What is the 'overlapping area'?

151: Please show all the data somewhere (appendix) so that the reader can see this.

153: change 'till' to 'until'.

158: please define 'operational system'.

161: remove comma and change to "aim *is* to predict".

162: I'm not sure what you mean by "no precursory deformation data is included". In line 123 you say that you give it 32 days of data and section 4.4 describes the different lengths of time used for training the models. Please clarify your meaning.

164: why 4 days?

176: Provide an overview of the numbers for us (absolute max, absolute mean, etc).

182: I don't see Wattenberg on Figure 1. Please include it.

183: You already describe the moving average. I don't think it needs to be repeated here.

206: I'm not sure what 'support the model' means.

207: Trial and error on what?

214: It also resembles the sm_profile data.

Section 4.3 I think here is where you should explain how neural networks work. Not above. And I suggest only including enough information for the reader to understand your model.

218: What do you mean disturb the model? You do this to keep the data at the same scale as the training data.

229: Please justify why you decided to share the memory rather than develop two individual models for the two benchmarks.

232: times series of deformation data, right? Please clarify.

234: I don't really see the value of this paragraph. Consider omitting.

291: But do all the examples have a strong driver? Why did the ones without a strong driver work well and yours did not? Do their study sites have different scales compared to this one?

Figure 8: please define the shaded background.

320: "*The* major challenge for the model…"

329: Please explain why reducing the number of parameters matters. Preventing over fitting, right. I think you say this at the end of section 2.2 but it was pretty convoluted. See general comment #5.

344: If the traditional least squares isn't tested why do you include it.

347: Can you be more specific on how you created this deformation rate model? Do you describe this somewhere that I missed?

Figure 11: I think you should consider merging figures 2,6, and 11. They show basically the same thing except some annotation.

424: I don't understand why this is the introductory sentence of this paragraph. Consider rewriting.

462: Just write out the meaning of SNR.

528: What assumptions? Please restate.

Figure 13: I don't see a reference to this figure anywhere in the text. Only the caption of figure 10. I think you should discuss this somewhere if you're going to include it in the main text.

558: Amplitude of what?

---

## Author Comment (AC2)

Rebuttal *Machine learning nowcasting of the Vögelsberg deep-seated landslide: why predicting slow deformation is not so easy*. [egusphere-2022-950]

We thank the reviewers for their detailed and thorough comments. We were pleased to read the reviewers appreciation on the quality of the work, as well as the recognition for the need to explore the limitations of machine learning.

Here we would like to address the concerns raised in the two reviews submitted during the discussion phase. Three comments that require a more elaborate discussion are discussed first, followed by comments that required clarification. Where possible the comments by both reviewers are merged and/or intertwined. Comments by Katy Burrows are shown in blue. Comments by the anonymous reviewer are shown in orange.

- **Environmental conditions**

  **2.** I'm a bit concerned by the environment predictor data you use in you model. The spatial resolution of the satellite data is often much coarser than the size of the landslide. I have a hard time seeing how these coarse resolution datasets could provide any meaningful information at the local scale you're looking at.

  **4.** Other studies (e.g., Thomas et al., 2019; Yatheendradas et al., 2019) have assessed the utility of satellite-based weather data for slope stability and found mixed results. I believe its possible that many of your issues could be attributed the poor representation of satellite data for your study site. I think this merits discussion or justification for why you think this is not an issue.

  Another thing that could adversely affect your model could be that small spatial scale rainfall events might not be captured by your satellite rainfall products, which have quite a coarse resolution (although unfortunately there would not really be any solution to this)

Satellite precipitation products are notoriously bad at measuring peak precipitation. However, unlike shallow landslides, deep-seated landslides are less sensitive to short periods of intense precipitation, illustrated by the 20-60 day lag time between precipitation and accelerated deformation (l. 147). A bias in the precipitation data, a consistent underestimation, for example, would be counteracted by the scaling in the neural network.

Thanks to the study by Pfeiffer et al. (2021) we know that the Vögelsberg landslide has this slow response (l. 147) and precipitation higher up in the catchment is relevant to the landslide as well (l. 526). Therefore, infiltration is a catchment wide rather than a slope limited process, approaching the lower resolution of the satellite precipitation measurements.

We agree that the spatial-temporal resolution of the satellite observations is not optimal. However, these products are operational globally, and the necessary foundation for an operational nowcasting system.

- **Deformation data**

  **3.** I also have concerns with the deformation data itself. If the data is so noisy, are you sure you can trust it at all? Are you certain that the deformation signal you are trying to replicate isn't an

artifact? If you are confident in the deformation signal, please better demonstrate why to the reader. For example, provide the accuracy of the data, the preprocessing step used by the Division of Geoinformation of Tirol, and elaborate on the corrections to the measurements. Based on the smoothing you conducted to get a usable signal, I find it difficult to trust this data. The poor quality deformation data may be partially responsible for your poor model fit.

In May 2016 a permanent automatic tracking total station (ATS; Leica TC1800) was installed on the opposite valley flank. Within the active landslide and in the surrounding area assumed stable, 53 retroreflecting prisms were installed on buildings and other man-made structures in a range between 600 and 1700 m. With this setup hourly measurements were conducted automatically, aggregated to daily means. However, the resulting displacement time series showed a drift towards north-west, owing to a minor movement of the installed ATS itself. This drift was removed assuming that reflectors at locations around the actively moving landslide did not move (no signs of damage owing to ground movement were observed). For each epoch a transformation matrix was derived by matching the coordinates of the respective measurements at stable locations to their initial positions at the beginning of the monitoring (arithmetic mean of first five measurements) based on the closed-form solution for rigid body transformation provided by Horn et al. (1988). By applying these transformation matrices to all measurements of each epoch, including the reflectors in the active landslide area, the observed pseudo-movements at stable reflectors were corrected in all displacement time series. The finally achieved accuracy of the displacement time series is in the order of ±0.54 cm/a (Pfeiffer et al. 2021).

The text of §3 (l. 148) will be expanded to reflect this.

- **Previous work (Table C.1)**

**5.** I don't think section 2 in necessary for this paper. I found it to be a bit burdensome and ancillary to the main point of the paper. Consider trimming it down to only describing the points that are pertinent to the model you use and then putting it in the methods and/or discussion where you describe the model(s) you use in this study.

**Table C1** This is a comprehensive table, but since not all the studies were included in the original table from Van Natijne (2020) there is no definition of some of the acronyms e.g. GRNNS.

**1.** You provide a table of several research articles that have produced now-cast models. In section 2, you also say that at least some of these articles were for deep-seated landslides. I think it would be valuable for you to explicitly highlight what was different between this study and the ones that seemed to have success with now casting. Do you know exactly why they had success and you didn't? You discuss the reservoir being a factor for some of these studies, which makes sense. However, did any other studies try to nowcast deep-seated landslides with success?

In section 2 we provided a concise introduction in now casting of deep-seated landslide focused on machine learning. This is, in our opinion, important to provide the reader a broader perspective of our work on landslide nowcasting and explain our findings later on. Except for one, all methods in the table are some form of machine learning. We did not look into the model or optimizer that was used as we did not compare results. We will remove the column with methods and this will also reduce the need for lists of acronyms and abbreviations. We prefer to keep the table, as it contains the expected references to earlier work. Most of these studies were on deep-seated landslides, that did not undergo catastrophic collapse. This will be added to the text and table caption.

All the cited papers that claim to be successful either had a strong driver (line 72), or split the signal in some way. In that case they typically subtract a trend from the data and apply a complex machine learning model to capture the smaller deviations from that trend. In Figure 10 we show that the mean/trend is already a very good predictor.

- **Minor comments**

  The title will be changed to "Machine learning *based* nowcasting […]".

  To support colorblind readers symbols were added to the color-coded lines in Figures 10 and 12.

  > 50: Please provide citations for this claim.

  This statement was not intended as a claim, but as a 'vision' for the paper. It will be reformulated as such.

  > 151: Please show all the data somewhere (appendix) so that the reader can see this.

  The data of all benchmarks is featured in Pfeiffer et al. (2021), Figure 3. A more explicit reference will be added to the text for the curious reader.

  > **Line 157-158** "Furthermore the amplitude of the filtered signal lags behind the original deformation signal" Why is this? Is it a side-effect of the filtering or have you done it deliberately?

  This is an unintended consequence of the filtering. This line will be removed to emphasize the previous, more important, statement that the onset of acceleration is severely dampened by this filter.

  > 162: I'm not sure what you mean by "no precursory deformation data is included". In line 123 you say that you give it 32 days of data and section 4.4 describes the different lengths of time used for training the models. Please clarify your meaning.

  Our model was designed to work in the absence of recent deformation measurements, and work on environmental conditions only. This sentence will be rewritten to clarify this.

  > 164: why 4 days?

  A four day prediction would demonstrate the model's ability to predict a tipping point based on the environmental conditions (acceleration, peak, deceleration). Furthermore, a four day prediction would give sufficient time for further investigation as part of an early warning system. This will be clarified in the text.

  > 176: Provide an overview of the numbers for us (absolute max, absolute mean, etc).

  We will, in addition to figure A.1, add the mean and maximum correlation to the text ($\mu_{|\rho|} = 0.16$, max $|\rho| = 0.7$).

  > 183: You already describe the moving average. I don't think it needs to be repeated here.

Agreed. However, in our perception the redundancy is desirable for consistency.

> 206: I'm not sure what 'support the model' means.

The deformation nowcasting model. This will be clarified in the text.

> **Line 209** Is API calculated from the ERA5 or GPM datasets? Does it make any difference which one you use?

Although better performance may be expected from the re-analysis in ERA5, the API time series is based on the GPM data, as this is an operational system that has data available with limited latency. This will be added to the text.

> 218: What do you mean disturb the model? You do this to keep the data at the same scale as the training data.

Indeed, the trained model will be sensitive to uncontrolled scaling of the input data. It will be clarified in text that the scaling of the data should be consistent with the scaling used during training.

> Section 4.3 I think here is where you should explain how neural networks work. Not above. And I suggest only including enough information for the reader to understand your model.

For the sake of brevity, we prefer to limit ourselves to giving a suitable reference here. We would propose to add Jian et al. (1996) as it is easily readable and complete on the workings and possibilities of such networks.

> 229: Please justify why you decided to share the memory rather than develop two individual models for the two benchmarks.

This was done to reduce the number of parameters. The later model lstm3-32 uses three memory cells, increasing the number of parameters required by a factor of three for the simplest combination of input parameters.

> 234: I don't really see the value of this paragraph. Consider omitting.

The reviewer is right that this paragraph is not clear. The conclusion, that from a parameter perspective a more information rich variable should be preferred over a more complex model, is missing and will be added to the text.

> 291: But do all the examples have a strong driver? Why did the ones without a strong driver work well and yours did not? Do their study sites have different scales compared to this one?

All studies in Table C.1 were conducted at slope level. Studies featuring multiple slopes often apply the same methodology to multiple slopes within the same catchment or along the same reservoir. We are unaware of regional studies that feature an automated analysis of many slopes with diverse drivers, as we mention in our outlook (l. 543).

> **Figure 6** The shaded areas for lines 1-4 begin before the deformation measurements. Is this a mistake?

This is not a mistake. The deformation training data is only available after the first set of 32-days of total station observations is available to the moving average filter. After that the model needs another 32-days to warm-up the memory in the LSTM-nodes. Theoretically these periods could have overlapped, at the cost of complicating the model.

This explanation will be elaborated on in the captions of figures 6 and 7, and the period will be marked in the figures as well.

> **Section 5**, **line 274-292** In the end, your model does not contain any snowmelt input, although you expected this to be relevant for the landslide. Could this be why your model only predicts the training data well in Summer and Autumn? Does a model including one of your snowmelt inputs (V3 or V4) predict Spring and Winter better (even if it's worse over all 4 seasons combined?)

This is an interesting hypothesis, that touches upon the information content of the variables, as well as the performance metric. However, evaluation of the performance over individual seasons would further reduce the length of the already short validation data set. We will gladly add this suggestion to §6.1.1.

> **Figure 7** Like my comment for figure 6, I wonder why the shaded area starts before your deformation dataset. Is this the "warm-up time" you describe in your figure caption? Or is that the part starting from early June 2016 where you have deformation data but no prediction? Maybe you could label this warm-up time on the time series with a box or shaded section?

Please find our reply under your previous question on Figure 6.

> 329: Please explain why reducing the number of parameters matters. Preventing over fitting, right. I think you say this at the end of section 2.2 but it was pretty convoluted. See general comment #5.

Agreed, the goal of the reduction of the number of parameters will be briefly restated in this section of the text.

> 344: If the traditional least squares isn't tested why do you include it.

Traditional least squares would be the obvious solution. Given the resemblance of the da-32 model to the least squares solution, it was mentioned for reference.

> 347: Can you be more specific on how you created this deformation rate model? Do you describe this somewhere that I missed?

The average deformation rate was calculated over the training period, and used as a prediction for the remainder of the time series. This mean squared error of this parameterless 'model' is shown in Figure 10. The text will be expanded to clarify this.

> **Section 6.1.1 Lines 360-365** Your $R^2$ value (0.31) seems low, but interpreting a single $R^2$ value is difficult. I'm not sure it's useful to include this metric when you have nothing to compare it to.

Agreed. We will remove this whole paragraph.

> **Section 6.1.2** I think I would have put this subsection in your methods section as it contains similar information to Sections 4.1 and 4.2

Indeed, the first paragraph replicates most of the considerations in §4.2 and has limited value for the discussion. We propose to remove this first paragraph of §6.1.2 and rename this section to 'derived variables'.

> **Lines 409-413** If you separate the two benchmarks in the model, would this result in them no longer being connected in space? (So one could accelerate independently of the other). I would have thought that since one part of a landslide moving is likely to destabilise another part, separating them would be a disadvantage.
>
> Actually, I think I have misunderstood what you mean to say in these lines, can you find another way to write this?

Although no spatial relationship is provided to our model, the relationship is found during training as two realizations of the same slope process, represented by the shared LSTM-node. This will be clarified in the text.

> Figure 11: I think you should consider merging figures 2,6, and 11. They show basically the same thing except some annotation.

Figure 2.6 shows the training periods used by the machine learning model. Figure 2.11 shows the accelerations periods as previously identified by Pfeiffer et al. (2021). The annotation is the main purpose of the figure, not the deformation time series. Therefore, no amplitude (y-axis) is shown in Figure 2.6. Figure 2.11 could be integrated into Figure 2.6, although the acceleration periods by Pfeiffer et al. (2021) are not introduce until §6.1. Therefore, we prefer not to merge the two figures.

> **Lines 480-486** I would specify Sentinel-1 since the temporal resolution of SAR satellites varies. Also, it is not clear here whether you are suggesting the use of InSAR as an input variable for the model, or if you are suggesting that maybe for other landslides where you don't have such detailed deformation data, deformation time series derived from InSAR could be used to train a similar model.

The first part of this paragraph is generic: InSAR may be used to train to train a similar model. The second part is specific to Vögelsberg and Sentinel-1. This will be clarified in the text.

> 424: I don't understand why this is the introductory sentence of this paragraph. Consider rewriting.

This sentence was a remnant of an old paragraph and will be removed.

> **Lines 522** For landcover changes as an input, won't you run into the same problem of temporal resolution as you found in the SAR data? And if your landcover product was derived from e.g. Sentinel-2, it could actually be worse because of cloud cover.

Indeed, the timing of fast changes can be difficult to capture by (optical) remote sensing. This reservation will be added in text.

> 528: What assumptions? Please restate.

The assumption that, given the low spatial resolution of the remote sensing data, a single pixel represents the conditions of the higher part of the catchment as well. This will be clarified in the text.

> **Lines 543-547** Here, with the EGMS product, it is based on Sentinel-1 data so you would only have a 12-day temporal resolution (Especially following the failure of Sentinel-1B), which would result in the same temporal resolution problem you discussed in Section 6.3.1

The temporal resolution between april 2016 and december 2021 was at the original 6-day interval. The current EGMS data set runs till the end of 2021, the same time as the failure of Sentinel-1B, and still has an (approximately) weekly frequency.

> Figure 13: I don't see a reference to this figure anywhere in the text. Only the caption of figure 10. I think you should discuss this somewhere if you're going to include it in the main text.

Indeed, the figure is superfluous in relation to Figure 10 and will be removed.

- **Textual corrections**

We thank both reviewers for their detailed textual corrections. These textual comments and minor comments will be addressed without further discussion.

---

## Editor Decision (ED1)

Dear Dr. van Natijne and co-authors,

Thank you for submitting your manuscript to our Special Issue on the use of Machine Learning in Natural Hazards Risk Assessment. We have received comments from two reviewers who commend the manuscript for its clarity and the account of the difficulties in pursuing Machine Learning for predicting slow deformation but would like to see major revisions made to ensure the manuscript can be accepted for publication. To be considered for future publication in this SI, the following points from the reviewers must be addressed, as you highlight in your initial response file.

- **Providing evidence to support justification in response to both reviewers' comments on the coarse resolution of environmental predictors.** Both reviewers pointed out that the coarse resolution of the environmental predictor data could be the cause for poor performance in the model used by the authors. The authors should provide evidence that the resolution is not too large for this problem and/or that the neural network would address these issues.
- **Justifying the use of the deformation data in this study.** Reviewer 2 points out concerns with trusting the deformation data that was used in this study. The authors should provide evidence as to why the deformation data are suitable for building an ML model.
- **Making a stronger connection to previous studies.** Both reviewers point out Table C.1 and other studies, desiring more detailed comparisons and references of the findings of this study with that of previous studies in the main text of the article (perhaps in the discussion).

**Sabine Loos, PhD**
Editor | NHESS Special Issue on Advances in machine learning for natural hazards risk assessment

---

## Author Response (AR2)

Minor revision of *Machine learning based nowcasting of the Vögelsberg deep-seated landslide: why predicting slow deformation is not so easy*. [egusphere-2022-950]

We were pleased to read the reviewer's appreciation on the improvements made to the manuscript. Here we would like to summarize the changes made based on the recommendations in the second review. Our improvements in the manuscript invoked by the comments, in brown, by the reviewer are detailed below. Independent of the reviewer's comments, the following textual changes were made to the manuscript:

- The colors of the lines indicating the deformation (ATS) in Figure 3 have been swapped to be consistent with the other figures in the manuscript.

- The ordering of the appendices, author contributions, competing interest statement, acknowledgments and references, as indicated by the editorial office.

- **Environmental conditions**

> **2.** I'm a bit concerned by the environment predictor data you use in you model. The spatial resolution of the satellite data is often much coarser than the size of the landslide. I have a hard time seeing how these coarse resolution datasets could provide any meaningful information at the local scale you're looking at.

We agree that the spatial-temporal resolution of the satellite observations is not optimal. However, these products are operational globally, and an important foundation for an operational nowcasting system. This is now explicitly mentioned in l. 59.

> **4.** Other studies (e.g., Thomas et al., 2019; Yatheendradas et al., 2019) have assessed the utility of satellite-based weather data for slope stability and found mixed results. I believe its possible that many of your issues could be attributed the poor representation of satellite data for your study site. I think this merits discussion or justification for why you think this is not an issue.

Satellite precipitation products are notoriously bad at measuring peak precipitation. The work by Thomas et al. (2019) and Yatheendradas et al. (2019), however, is on shallow landslides, that are more sensitive to this type of precipitation. We have added references to their work and a brief hint on the differences on l. 28.

> Additional point building on points 2 and 4 above. Since you have local precipitation measurements and a detailed hydrological model from Pfeiffer et al., 2021, I think it would be worth while to compare the satellite data to these more detailed data to see if that could be a major factor in poor model performance. As you say in the paper, "garbage in, garbage out".

Although this would be an interesting experiment we considered it to be out of the scope of this study. We now hint on this possibility and have made it explicit that we consider this out of scope of our study on l. 492. Our satellite focus is highlighted in all sections of the paper: in the abstract (l. 6, 12), introduction (l. 55), model variable selection (e.g., l. 189), and discussion (e.g., l. 502), and conclusion (l. 581). We changed l. 185 to further emphasize our desire for a satellite focus even though local deformation data is used in this study.

- **Previous work (Table C.1)**

> **1.** You provide a table of several research articles that have produced now-cast models. In section 2, you also say that at least some of these articles were for deep-seated landslides. I think it would be valuable for you to explicitly highlight what was different between this study and the ones that seemed to have success with now casting. Do you know exactly why they had success and you didn't? You discuss the reservoir being a factor for some of these studies, which makes sense. However, did any other studies try to nowcast deep-seated landslides with success?

> Additional comment on this. In your response letter, you say that the manuscripts you include in table C1 that don't have a strong driver only try and estimate smaller deviations from the trend. Does this suggest that they have significantly more data to develop and validate their models? Could this not be a major difference in your approach worth addressing explicitly? I'm not an expert on neural networks, but my impression is that they require much more data than alternative machine learning approaches. Using only 1500 data points to estimate ~68 parameters seems very problematic. I think you were getting at this in section 6.2.1, but a more explicit discussion is warranted.

To emphasize the dependency of many studies on a reservoir level, the references to previous work in Table C.1 has been split into two sections, with and without the reservoir level as input variable to the prediction. For readability, no 'track changes' version of the table and caption is provided.

Furthermore, we have expanded the text on this in l. 71 – 81. We now provide examples as well expand on our standpoint regarding trend splitting.

- **Minor comments and textual corrections**

> 23: Change 'global' to 'globally'

> 74: The last comma on this line needs to be deleted.

> 129: don't delete 'in'.

These textual corrections have been applied.

> 154: Their figure 3, right? Please clarify.

Indeed. This has been clarified.

> 158: Could you briefly mention what rainfall data Pfeiffer et al. Used?

This has been clarified as 'in-situ observations'. For further details on the full collection of local observations available the readers are referred to cited work.

> 199: Why 32 days? Move your explanation below up here.

There is no mention of 32 days in l. 199, this comment might reflect on l. 190 instead? The minimum filter length to remove all negative deformation rates induced by the noise was 32 days. This is explained before, in l. 162. The 32 day period was repeated for convenience here.

> Table 1: Consider defining API before this table or in the table.

Indeed, the API is introduced directly after rather than before the table. However, we prefer to provide an overview of the variables first. The API (V10) and seasonal noise (V11) require further explanation that would distract from the bigger picture of hydrometeorological variables.

> 224: What expert?

This has been made explicit as 'experienced landslide hydrologist'.

> Figure 12: I would explicitly say the pink line is the mean deformation rate.

Adapted. Furthermore, the caption has been changed to clarify that this line indicates the mean of 1, 2 or 3 years of deformation.

---

## Author Response (AR3)

Dear Editors,

As requested by the editorial staff all figures have been tested for their accessibility to colorblind readers. Figure 4 has been improved to better indicate the difference between the reference signal and the random sample.

Kind regards,
On behalf of the authors,

Adriaan van Natijne